# What are we learning with Yoga? Mapping the scientific literature on Yoga using a vector-text-mining approach

Rosangela Ieger-Raittz[1], Camilla Reginatto De Pierri[2,3], Camila Pereira Perico[2,4,5], Flavia de Fatima Costa[2,4,5], Elisa Garbin Bana[2,4,5], Leonardo Vicenzi[2,4,5], Diogo de Jesus Soares Machado[2,4,5], Jeroniza Nunes Marchaukoski[2,4,5], Roberto Tadeu Raittz[2,4,5]*

**1** Graduate Program in Physical Exercise Medicine in Health Promotion, Health Sciences Sector, Federal University of Paraná, Curitiba, Paraná, Brazil, **2** Laboratory of Artificial Intelligence Applied to Bioinformatics, SEPT, Federal University of Paraná, Curitiba, Paraná, Brazil, **3** Department of Biochemistry and Molecular Biology, Federal University of Paraná, Curitiba, Paraná, Brazil, **4** Graduate Program in Bioinformatics, SEPT, Federal University of Paraná, Curitiba, Paraná, Brazil, **5** Associate Graduate Program in Bioinformatics, SEPT, Federal University of Paraná, Curitiba, Paraná, Brazil

* raittz@ufpr.br

## Abstract

The techniques used in yoga have roots in traditions that precede modern science. Research shows that yoga enhances quality of life and well-being, positively impacting physical and mental health. As yoga gains acceptance in Western countries, scientific studies on the subject increase exponentially. However, many of these studies are considered inconsistent due to the diverse methodologies and focuses in the field, which creates challenges for researchers and hampers progress. This study aims to develop a comprehensive framework for existing literature on yoga, facilitating multidisciplinary collaboration and bringing new light to relevant aspects. Given the complexity of the subject, advanced modeling techniques are necessary. Contemporary artificial intelligence methods have advanced Bioinformatics, including text mining (TM), allowing us to employ vector representations of texts to derive semantic insights and organize literature effectively. Based on TM resources, we provided a better general understanding of yoga and highlighted the relationships between yoga practice and various domains, including biochemical parameters and neuroscience. It also reveals that practitioners can learn to engage with their bodies and environments actively, enhancing their quality of life. However, there is a lack of research exploring the mechanisms behind this learning and its potential for further enhancement. Vector TM has made it possible to bolster and improve human analysis. The set of resources developed allowed us to determine the mapping of the literature, the analysis of which revealed 4 dimensions (exercise, physiology, theory and therapeutic) divided into 9 cohesive groups, representing the trends in the literature. The resulting platforms are available to Yoga researchers to evaluate our findings and make their forays into the existing literature.

**Data availability statement:** The data obtained from the analyses performed in this study are available at https://aibialab.github.io/HTMLTM_Yoga and the supplementary material (files S1-S12) deposited in Zenodo repository at https://zenodo.org/uploads/14763946.

**Funding:** Brazilian National Council for Scientific and Technological Development - CNPq (grant number 440412/2022-6) granted for publication fees. The Coordination for the Improvement of Higher Education Personnel - Brazil (CAPES) awarded to CPP and DJSM also partially supported the works. The funders had no role in study design, data collection and analysis, publication decisions, or manuscript preparation.

**Competing interests:** The authors have declared that no competing interests exist.

## 1. Introduction

Yoga practice (YP) has cultivated a profound cultural legacy, with its teachings spanning thousands of years [1–3]. However, merging traditional wisdom with scientific principles poses a significant challenge, particularly because many yoga experts are not associated with the academic community [4]. The scientific literature on yoga encompasses thousands of articles and numerous reviews. The benefits reported in this extensive body of work prompted further exploration of innovative applications, and the trend suggests that interest in this subject will continue to expand.

Research on yoga covers a wide range of fields, from biochemistry and physiology to psychiatry and the social sciences, for instance. In contrast, a lack of consensus on general guidelines for multidisciplinary approaches hinders the advancement of scientific knowledge, as research findings are often under-analyzed due to difficulties in linking disparate results [4]. The heterogeneity of experimental designs and the absence of a standardized approach to Yoga diminishes confidence in the outcomes, as comparable results are difficult to achieve. To overcome this challenge, the term Yoga-Based Practice should be adopted in generic research [5], to unify various studies across different yoga styles.

Recent studies offer in-depth discussions on the relevance of YP to various clinical conditions, including its mechanisms of action and therapeutic effects [6,7]. In this setting, we identified research centered on the construction, discussion, and reformulation of frameworks aimed at integrating these insights [8,9]. Many of these models strive to connect traditional approaches with scientific arguments [10]. Other strategies target specific technical aspects of a particular area of knowledge [11,12]. Collectively, these theoretical studies serve as a valuable resource for understanding Yoga as a research subject.

Emerging technologies in information, particularly Artificial Intelligence and Data Science, can effectively illuminate and help to interpret data, revealing hidden meanings and intricate relationships between concepts [13,14]. Despite their potential, our research indicates a notable lack of studies applying Data Science to the analysis of yoga literature. We identified only one study focused on future research trends in yoga, alongside some recent works exploring connections to the nervous system [2]. Moreover, other investigations have utilized advanced language models like Bidirecional Encoder Representations from Transformers (BERT) to provide topic modeling on key texts such as the Upanishads and the Bhagavad Gita [15]. Furthermore, text mining (TM) techniques, particularly those using vector representation, have shown significant promise in representing textual documents [16–18]. This method involves representing text-associated vectors as points in a multi-dimensional space, allowing for direct comparisons between documents through geometric distance measures [19].

This study presents a novel text-mining environment that employs vector representation approaches to analyze Yoga literature. Through TM technology, we synthesized existing literature to uncover consistent structural elements and relationships among various aspects discussed in the articles. Our investigation aims to address several fundamental questions: 1) Does Yoga possess a robust

scientific foundation? 2) Can the effects of Yoga practice be generalized to other fields, particularly health? 3) Are there underexplored topics within Yoga practice that warrant deeper analysis? 4) How can researchers contextualize themselves within yoga literature and its subfields? Our analysis reveals that a clear articulation of the literature offers valuable insights into the potential benefits of yoga practice (YP) and the profiles of its practitioners. Additionally, we examined the potential evolution of theoretical frameworks to instrumentalize yoga researchers, notably by incorporating learning measures as key outcomes of yoga practice.

## 1.1. Definitions and technical details

The vector representation of texts has already been used in the literature [15] and has proved advantageous in allowing various subsequent analyses, including machine learning methods. To vectorize documents, we propose using a technique that vectorizes biological sequences (for example, DNA sequences) into numerical vectors, the SWeeP tool (see section 1.1.1). Using a "trick," we can convert texts into biological sequences, done by the Biotext tool (see section 1.1.2), and then vectorize them with SWeeP. We can apply analysis and visualization methods from the numerical vectors representing these texts to evaluate the relationship between these documents through their content and between the present terms. The terms contained in the documents can be considered individually or using composition logical expressions (LOGEXP) (section 1.1.3.1). We used the phylogenetic tree to visualize the relationships between the terms and LOGEXP (section 1.1.3.2). Clustering (section 1.1.3.3) groups the vectors (documents or terms) based on their content. The Principal Component Analysis (PCA) allows easy visualization of the vector data distribution by highlighting its most relevant components. T-distributed Stochastic Neighbor Embedding (tSNE) is a non-linear method that allows the visualization of the complexity of vectors in two dimensions.

### 1.1.1. Spaced words projection - SWeeP.
The SWeeP method was initially developed to represent large datasets of biological sequences, such as genomes and proteomes, in compact vectors [20]. The technique uses spaced-words to index subsequences, creating a high-dimensional vector that is then projected onto a reduced-dimensional by a random orthonormal basis. This process allows efficient sequence comparison, even on a large scale, as demonstrated by constructing phylogenetic trees of mitochondria, bacterial genomes, and viruses [20–22]. The main advantage of SWeeP lies in its superior speed compared to traditional sequence alignment methods and other alignment-free techniques, without compromising the quality of the analysis. Furthermore, vector representation facilitates the application of machine learning and principal component analysis techniques for knowledge discovery in biological data [20].

### 1.1.2. Biotext.
BIOTEXT is a software package that allows the application of bioinformatics tools for text mining. The central idea is to convert texts into biological sequence format (BSF) using two main functions: AMINOcode, which replaces text characters with letters representing amino acids, and DNAbits, which converts text characters into DNA sequences. The study demonstrates some applications of BIOTEXT in analyzing texts and highlights its efficiency in handling a large volume of data. Also presented is the ability to perform text mining tasks such as information retrieval (IR), named entity recognition (NER), and information extraction (IE) [23].

### 1.1.3. Technical details.
1.1.3.1. *Logical expressions:* Logical expressions (LOGEXP) are widely used in mathematics, computing, and natural language. LOGEXPs are made up of logical operators and allow textual sentences to be evaluated for the presence of specific textual patterns. In this study, these operators are essential tools for filtering texts by their content in the search for particular patterns (LOGEXP). There are several logical operators, but the three main types are 'AND,' 'OR' and 'NOT.' To illustrate these operators, consider the text below:

*"yoga as a therapeutic intervention for adults with acute and chronic health conditions. Objectives: overview of the quality, direction, and characteristics of yoga interventions for the treatment of acute and chronic health conditions in adult populations.[...]"*

(1) conjunction (AND): represented by '&', returns true only if both propositions are positive.

- yoga AND health: TRUE (the text contains both terms)

- yoga AND injury: FALSE (contains only 'yoga' term)

(2) disjunction (OR): represented by '|', returns true if one or both sentences return positive.

- injury OR chronic: TRUE (the text contains the 'chronic' term)

- treatment OR chronic: TRUE (contains both terms)

- injury OR cold: FALSE (contains none of the terms)

(3) negation (NOT): represented by '~', returns true if the sentence returns negative.

- NOT injury: TRUE (does not contain 'injury')

- NOT yoga: FALSE (the text contains the 'yoga' term)

    We can also combine logic operations: '{}' are used to separate each operation.

- {chronic OR acute} AND conditions: TRUE (contains both, "chronic & conditions" and "acute & conditions")

*1.1.3.2. HyperText Markup Language:* HyperText Markup Language (HTML) is a markup language widely used to define the structure and content of web pages. It is a structured language based on tags, possibly including text, images, audio, and video, and easy navigation through hyperlinks [24]. HTML can be used as a visualization tool for accessing and navigating text mining results due to its practicality and interactivity.

*1.1.3.3. Phylogenetic Trees and Clustering:* Dendrograms are diagrammatic constructions that make finding relationships between study objects possible. In hierarchical clustering, the dendrogram represents the correspondence between the clusters obtained. In phylogenetic analysis, dendrograms are phylogenetic trees representing the evolutionary relationships between organisms with common ancestry. This representation comprises nodes (links between the branches) and branches (lines), as in the diagram of Fig 1. Closer branches are more closely related than distant ones.

    This representation is widely used in biology and bioinformatics, and various algorithms for constructing trees have been developed, such as Unweighted Pair Group Method using Arithmetic averages (UPGMA) [25] and Neighbour Joining (NJ) [26]. When applied to natural language, the relationship between the terms in is represented by the proximity of the branches. In this way, the tree reflects the grouping of points (terms) and reveals their relationships.

    Clustering methods aim to group data by similarity. Various techniques have been developed (partitioning, hierarchical, etc.) with variations in the clustering criteria applicable to multiple disciplines. Hierarchical clustering aims to hierarchize data to obtain a dendrogram-type relationship [27].

## 2. Methodology

### 2.1. Database

We focused on articles with the YOGA theme available in the PubMed database. Articles were selected if they included the word "YOGA" in the title or abstract and were published between January 1, 1970, and June 3, 2022, totaling 6,905 articles containing the following fields: PMID (PubMed identifier), Title, Abstract, Authors, and Data. Exclusion criteria included articles with empty title or abstract fields, titles plus abstracts containing fewer than 300 characters, and articles with duplicate titles. After these filtering steps, we obtained a set of 5,782 articles (called TextDocAll) that include the word

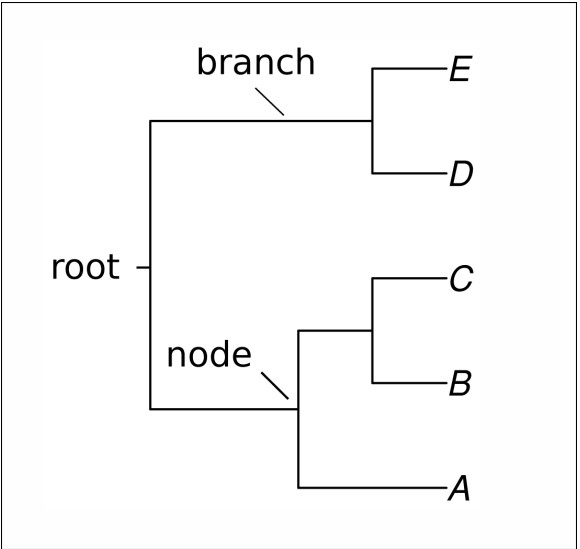

**Fig 1. A simple representation of a phylogenetic tree.** Each letter (A-E) represents a leaf, while the lines depict branches. The points where branches meet are called nodes, and the most basal node in the tree is referred to as the root.

"YOGA" in the title or abstract. This preliminary set provided us with an initial overview of the literature. It allowed us to verify that articles with the term "YOGA" only in the abstract do not have yoga as their primary object of study and, therefore, were discarded. As a final filtering stage, only articles with "YOGA" in the title were selected, making up the TextDoc set of 2,929 articles.

A text document consists of characters concatenated with a title and its respective abstract. We will refer to the corpus in the complete TM environment as TextDocAll and in the specific analysis TextDoc. Additionally, metadata related to each document was preserved for complementary access. A schematic diagram of the methodology is illustrated in Fig 2.

### 2.2. Documents and words vectors

To create the vector systems, we employed a strategy for semantic analysis that entailed transforming TextDoc into a biological information format, thereby enabling the exploitation of Bioinformatics tools. First, we constructed base vectors (Wbase). The projection length for SWeeP vectors was set to 1,369, the same as that used to vectorize whole genome sequences [21].

The word embedding adopted defines vectors representing a specific word (Wwrd) as a mean of the Wbase of TextDoc containing such a word (Eq 1).

$$Wwrd_i = mean\left(Wbase_j\right),\ j\ are\ documents\ containing\ Word_i \tag{1}$$

The final vectors representing the TextDoc documents (Wtxt) in the corpus are the average of Wwrd for the words contained in each document (Eq 2).

$$Wtxt_k = mean\left(Wwrd_q\right),\ q\ are\ words\ contained\ in\ Wtxt_k \tag{2}$$

In the proposed model, words and documents are dots in the same vector space where any word is comparable to another word or document (and vice versa) through vector distance metrics. We chose a subset of words (WRD) to

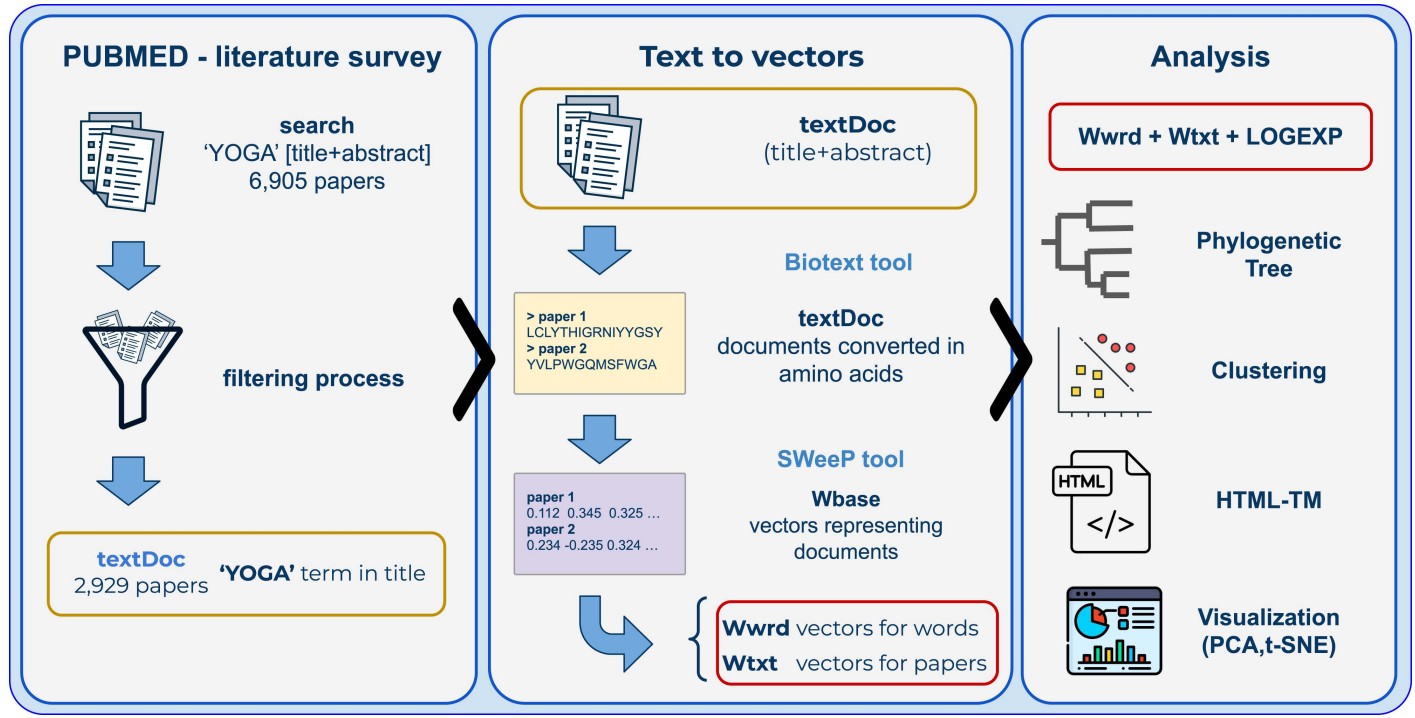

**Fig 2. Schematic of the methodology.** The first stage consisted of surveying the literature on PubMed. Only texts with 'YOGA' in the title were filtered for the subsequent stages. The texts (title and abstract) were processed, and a vector representing each word, document, and logical expression was obtained. This representation permitted phylogenies, clustering, visualizations, the HTML-Text Mining (HTML-TM), and subsequent analyses.

compose the relevant words list, considering the relative frequency of the words in the YOGA PubMed search to the Kaggle frequency [28]. The vectors (Wbase, Wwrd, Wtxt, and Wlog – Tables S1, S2, S3, and S4, respectively) and correspondent lists of words (WRD – Table S5), texts (TextDoc – Table S6) and logical expressions (Nlog – Table S7) are available in the repository Zenodo (https://doi.org/10.5281/zenodo.14763946).

### 2.3. Semantic search using vectors

Through vector distance comparison, this structure permits determining which words are closest to a given query (other words, papers, or LOGEXP) in texts or word databases. In these articles, an uppercase word, or a list of words (or text codes) separated by a hyphen, represents a query for a semantic search whose corresponding vector is calculated by the average of vectors to be searched.

For instance, NEUROLOGICAL-BRAIN-CORTEX is a query WQ – to be searched in Wwrd0, Wwrd, or Wtxt (Eq 3).

$$WQ = (Wwrs('NEUROLOGICAL') + Wwrd('BRAIN') + Wwrd('CORTEX'))/3 \qquad (3)$$

For some tasks, we used a smaller set of PCA (principal component analysis) components (50-100-300) instead of the whole vector when more generalization was needed.

### 2.4. String searches and logic expressions

We define logic expressions (LOGEXP) for this study to represent the examples involved with string searches. Operators utilized are: '(', ')' (segmentation), '|' (or), '&' (and), '~' (not), and '_' (a space is necessary). The parameters are substrings

that may occur in a TextDoc or a Document Title. In some dendrograms (phylogenetic trees) in complementary material, the parenthesis may appear replaced by braces due to the visualization tool's restriction. For instance, a vector representing the meaning of documents containing the substrings ' CARDI' ('CARDI' with space before) and 'VASCUL' is defined by ('_CARDI & VASCUL'). In the example, the substring ' CARDI' must be preceded by a space to validate this match.

## 2.5. HTML with Text Mining for YOGA

For every word in the set of top related words WRD (n = 2,878), we searched and saved the nearest 30 correlated terms from the complete list of words (n = 20,052) and 30 nearest documents from the list of documents TextDoc (n = 2,929). We also calculated an index of the year usage for each word.

We create two HTML-TM (Hyper Text Markup Language with Text Mining) files that allow users to exploit the text mining results on Yoga literature without deep programming or AI skills. The first is a file containing a short list of the seven best hits for each word in WRD, a phylogenetic tree of the 30 top related terms, and the list of 30 closest papers in TextDoc (WORDS.html) to represent the word's relationships. The second HTML-TM file (TEXTS.html) allows users to search for similar articles. The paper coding numbers in both files are consistent, facilitating quick navigation of the literature contents, accessible at the site https://aibialab.github.io/HTMLTM_Yoga.

## 2.6. Data Visualization and Programming

We constructed all Hierarchical clusters with the Neighbor-Joining method [26] for phylogenetic trees. The tools Dendroscope [29] and ITOL [30] are used for tree visualization and presentation. The Euclidean distance is the metric used for distance calculation in the LOGEXP tree; Lk-norm (k = 0.3) for trees in HTML-TM [19]. t-SNE [31] dimension reduction was applied for the 2-dimension visualization of vectors in examples.

Regarding programming, the entire TM approach was developed in-house and adapted for this study. The supporting tasks were programmed mainly in MATLAB [32], including constructing the HTML-TM files. The code developed for text processing and HTML construction, along with the rest of the supplementary material, is available on the Zenodo platform (file S11).

## 3. Results

### 3.1. Data analysis

The final files after the analysis had the following results: words (n = 2,878), texts (n = 2,929), LOGEXP (n = 113), available in supplementary files 'WRD', 'TEXT' and 'LOGEXP' (S5, S6, and S7 Tables, respectively). The phylogenetic tree with all words in the WRD set is available in the supplementary (S8 Fig). Studying this diagram, we performed our initial analysis and got acquainted with terms relations and their global disposition. These previous interactions oriented us for the first exploratory incursion into Yoga literature. We covered all the review articles available (n = 306) to define a prospecting list of LOGEXP (n = 113) (S10 Fig). Despite directly clustering the articles, we chose this strategy to derive a more meaningful structure for a multidisciplinary Yoga researcher. We built a vector from each resulting LOGEXP according to the same role used to embed the words, allowing us to exploit them in semantic searches. The phylogenetic tree rooted in the term 'YOGA' of the LOGEXP is in Fig 3. For the name of the leaves, we appended to the LOGEXP text the numbers of hits it brought from TextDoc and TextTitle (title for each abstract in TextDoc), respectively. The union of all searches with LOGEXP recovers 99.86% of the documents and 90.58% of the titles.

We employed a proprietary TM toolset to adapt resources to our purposes and tune vector configurations freely. The application of AI generated resources that supported human interpretation of the yoga literature, uncovering the structures already present in it. The interactive human-model relationship shaped these structures, and the role of each developed resource is detailed in Table S12, presented in order of execution. Fig 3 shows an intuitive example of comparing three

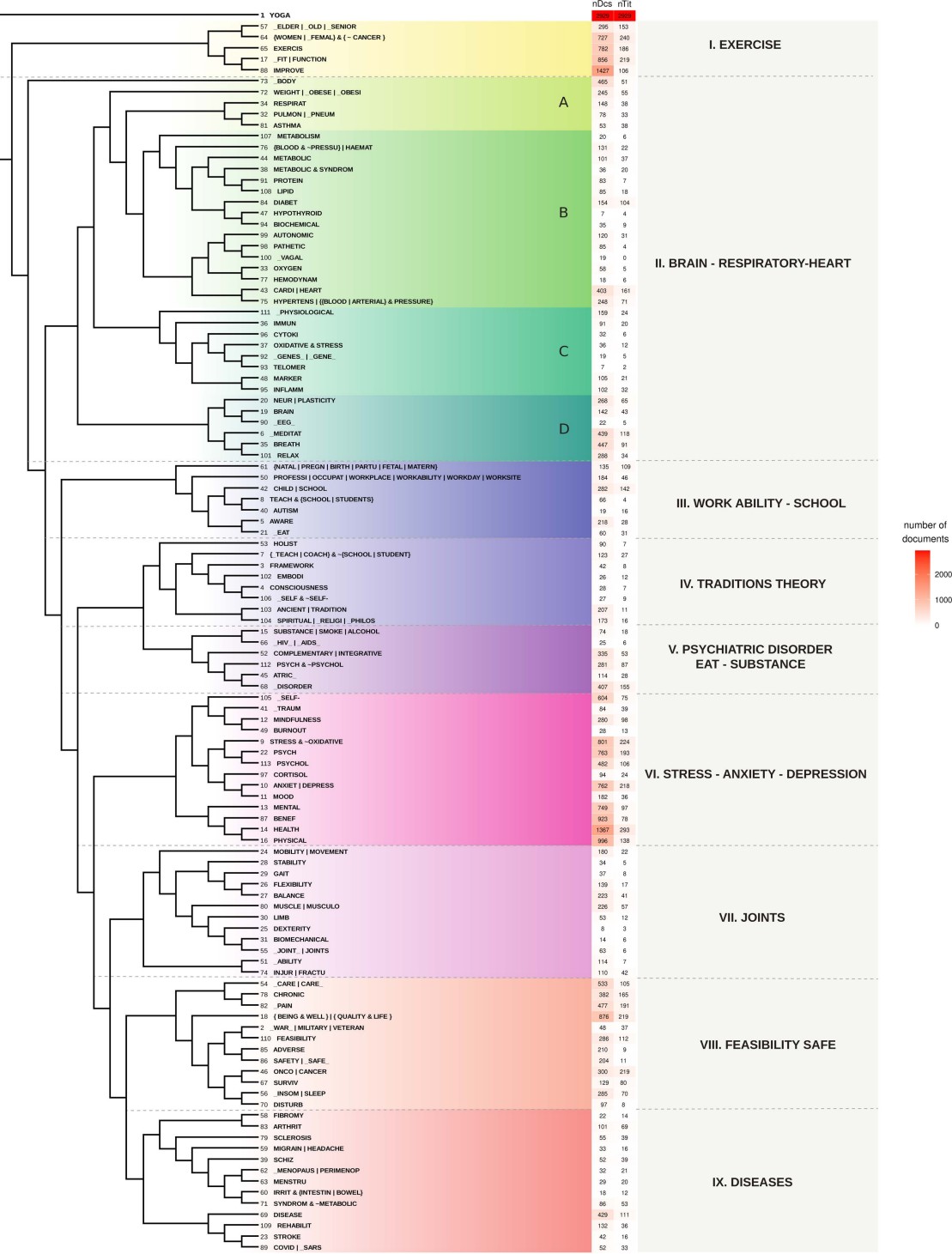

**Fig 3. Mapping of Yoga literature with all 113 logical expressions (LOGEXP) studied.** The phylogenetic tree presents the relationship of the 113 LOGEXP and divides the findings into nine groups with related topics. The tree characterizes the field of yoga as a reflection of the literature. Each branch presents a LOGEXP and its identification number. It also indicates the number of studies/documents (nDcs) and the number of titles (nTit) hit by the logical expression. Group II.BRAIN-RESPIRATORY-HEART is subdivided into four groups: A. Respiratory, B. Metabolic and Cardiovascular, C. Biomarkers, D. Neurological.

searches by related words in the context of all words (WRD0). We explored the "semantic power" proposed by vector representation to create a structured synthesis mapping of the literature based on the LOGEXP. The results derive mainly from the

studies with 'Yoga' in the title since the other studies did not focus on the practice of

yoga. We also projected a no-cost way to make these resources available to Yoga researchers in HTML-TM preprocessed files. The HTML structure made navigating and exploring the literature easier, allowing us to investigate the associations initially raised with the phylogenetic trees. This foray into the literature resulted in the literature mapping described in detail in the subsequent sections.

The HTML-TM environment will allow users to access this paper's results and perform semantic searches in the knowledge base. Phylogenetic trees, vectors, and corresponding generating data are available and described in more detail in the Supplement. A tutorial for using HTML-TM is also provided to make it easier for the researcher to consult the content. A reader interested in vector representation or word embedding can find vectors in supplementary tables (Tables S1, S2, S3, and S4).

## 3.2. Literature mapping based on LOGEXP

Each LOGEXP, like for words, is associated with a vector that can be searched or compared, making it possible to cluster corresponding vectors (Fig 3). The tree's root is the term YOGA, which is the most representative and occurs in all the documents. It determines that the first branches are the most generic, while the branches towards the end are more specific. As the LOGEXP set was chosen to represent all the literature, we used the diagram to understand the distribution of the articles in the more representative subjects. We identified nine cohesive topics and enumerated them (I-IX), attributing a name based on some LOGEXP within each topic. These topics form hierarchically related groups, which we interpreted as the four principal dimensions of the yoga literature organization. From the general to the specific, we propose the following dimensions: Yoga as a physical and mental exercise (I); Yoga affects the body and physiology (II); Psychosocial effects and theory of Yoga (III-VI); and The therapeutic contribution of Yoga (VII-IX) described in Table 1. It is important to note that neither the nine groups nor the four dimensions are necessarily mutually exclusive. Instead, they are interrelated and can be viewed as characteristics of the literature rather than distinct groups of articles. Fig 4 helps to visualise the overlap between the groups detailed in Table 1. However, our objective is to analyze these topics by examining the most relevant results (articles) for semantic search in an organized manner based on the distance from the vector to the centroid of each topic. This approach aims to connect the actual content of the papers and complement them with other investigations into the remaining literature when necessary.

### 3.2.1. Yoga as a physical and mental exercise.
This dimension, characterised by the group I - Exercise, comprises five distinct LOGEXP, totalling 4,087 article hits and 904 title hits (Table 1) – note that the articles and titles present in a dimension may also be present in other dimensions, so the total number of hits can exceed the total of documents. The LOGEXP with the highest number of articles is "IMPROVE" (LOGEXP 88), while the LOGEXP with the highest number of titles is "{WOMEN | _FEMAL} & {~CANCER}" (LOGEXP 64, Fig 3). Yoga is considered a light to moderate impact physical activity. Due to its characteristics, it facilitates a mindful approach, so it is regarded as a safe form of exercise and rehabilitation for the body systems. Furthermore, it is viewed as a practice that promotes physical, emotional, and neurological benefits, enhancing well-being and quality of life.

This literature reveals that yoga has been shown to promote health outcomes compared to other physical activities or no physical activity over time. Studies frequently involve people to improve their bodily functions. These studies include older adults, obese individuals, and individuals recovering from surgeries, traumas, and non-communicable diseases. Most participants in studies related to gender are female. Significant improvements have been observed in healthy older people's quality of life and physical measures [35]. Several systematic studies have compared YP to conventional physical exercise; for instance, [33] conducted a comparative analysis of yoga practice and aerobic exercise to reduce anxiety symptoms. The findings indicated that yoga practice was more effective in this regard.

**Table 1.** *Summary of results generated by mapping based on logical expressions (LOGEXP).*

| Dimensions | Topic (Groups) | Sub-groups | Description | nLOG-EXP | nDcs | nTit | Unq | Cov |
|---|---|---|---|---|---|---|---|---|
| Yoga as a physical and mental exercise | I - Exercise | | Approaches yoga as a combination of physical exercise and meditative practices, and it has physical, emotional, and neurological benefits. It has positive effects particularly on the elderly, the obese, and those with chronic illnesses. | 5 | 4,087 | 904 | 2,251 | 0.77 |
| Yoga affects the body and physiology | II - Brain/ Respiratory/ Heart* | A. Respiratory | Breathing techniques used in meditation and relaxation have an effect on the regulation of the autonomic nervous system (ANS) and cardiovascular system. Yoga Practice (YP) plays a role in improving lung function and reducing symptoms related to asthma and other conditions. | 6 | 989 | 215 | 776 | 0.26 |
| | | B. Metabolic and Cardiovascular | Benefits in controlling risk factors for heart disease, hypertension, obesity and diabetes. Effects of YP on HRV, the balance between sympathetic and parasympathetic activity, improved haemodynamic function and cardiovascular adaptation. | 16 | 1,603 | 505 | 768 | 0.26 |
| | | C. Biomarkers | Various biomarkers assess the physical and biochemical effects. YP has effects on the reduction of inflammatory markers, improvements in the regulation of the immune system and indications of a positive influence on cellular aging, oxidative stress and neuroplasticity. | 8 | 551 | 122 | 368 | 0.13 |
| | | D. Neurological | Yoga is associated with changes in brain activity. The literature suggests that the practice regulates the ANS, reducing stress and impacting cognitive functions. | 6 | 1,606 | 356 | 1,019 | 0.35 |
| Psychosocial effects and theory of Yoga | III - Work ability/ school | | Impacts of Yoga and Mindfulness in educational and professional contexts. Evaluation of the benefits of YP on stress, anxiety, depression, self-esteem, academic performance, socialisation and self-regulation and self-perception skills. YP helps with emotional regulation in students and communication in children on the autistic spectrum. | 7 | 964 | 376 | 784 | 0.27 |
| | IV - Traditions theory | | It addresses the scientific, philosophical and socio-economic aspects of yoga. It explores the practice in social contexts, work environments and interpersonal relationships. Yoga is described by three fundamental components: 'POSTURES', 'ATTENTION' and 'RESPIRATION'. | 8 | 716 | 97 | 563 | 0.19 |
| | V - Psychiatric disorder Eat/ Substance | | Yoga as a complementary therapy for psychiatric disorders, including eating disorders and addictions. Studies indicate a correlation between yoga and psychiatric biomarkers, suggesting new therapeutic applications and the potential to integrate yoga into psychiatric treatment. | 6 | 1,236 | 347 | 929 | 0.32 |
| | VI - Stress/ Anxiety/ Depression | | Yoga acts as a therapeutic complement in psychology for mood disorders, stress and trauma. YP's relationship with cortisol reduction and improvements in stress management and mood disorders. Mindfulness as a state of mind and as an associated technique. | 14 | 8,115 | 1,632 | 2,568 | 0.88 |
| The therapeutic contribution of Yoga | VII - Joints | | YP improves mobility, balance (static and dynamic) and gait, with applications in neuromuscular rehabilitation and injury recovery. Studies have shown benefits for osteoarthritis, diabetic peripheral neuropathy and spinal cord injuries. | 12 | 1,201 | 226 | 802 | 0.27 |
| | VIII - Feasibility safe | | Studies focus on patients with cancer, chronic pain and other clinical conditions, and analyse the safety, adherence and effectiveness of yoga as a complementary therapy. Yoga can improve sleep quality, mood, stress, fatigue and physical and social functionality. Studies | 12 | 3,827 | 1,226 | 1,821 | 0.62 |
| | IX - Diseases | | Yoga helps in the safe and alternative rehabilitation of neurological (stroke, Parkinson's, multiple sclerosis) and cardiovascular diseases. Benefits include improved cognition, mood, heart health, balance and pain reduction in osteoarthritis, fibromyalgia and migraine. Yoga can help treat IBS, dysmenorrhoea and post-menopausal osteoporosis. | 13 | 1,083 | 479 | 841 | 0.29 |

*Note*: **nTit**: Number of Titles; **nDcs**: Number of Document hits recalled by the LOGEXP whole set in a dimension; **nLOGEXP:** Number of LOGEXP in the topics. **Unq:** unique: number of different articles per group**; Cov:** Coverage: number of uniques in the group divided by the total number of articles * For group II as a whole we have: nLOGEXP: 35, nDcs: 4,289, nTits: 988, Unq.:1,824, Cov.: 0.62.

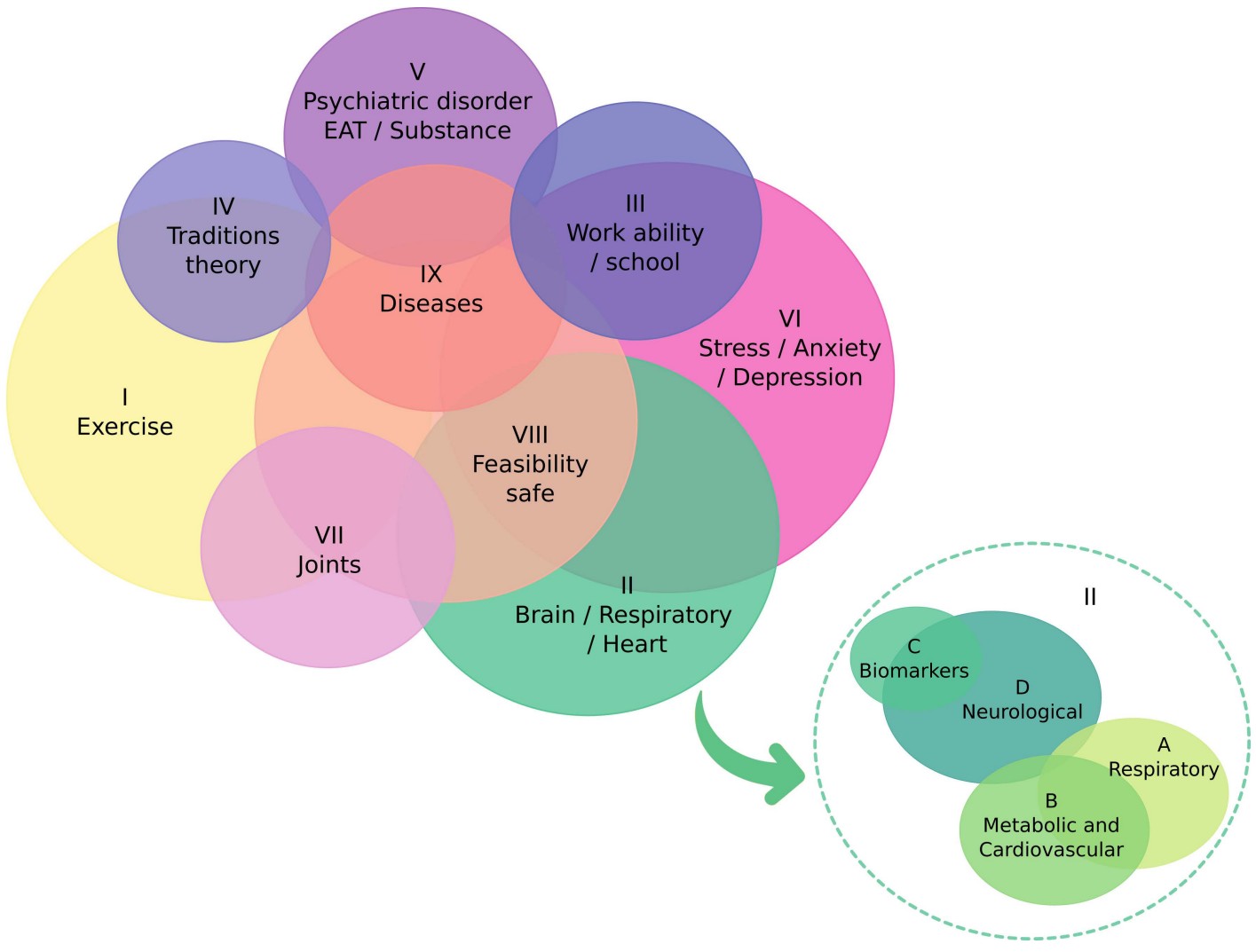

**Fig 4. The circles correspond to the groups in Table 1. We vectorized each group by the mean of logical expressions (LOGEXP) vectors inside it.** To define the centers of the circles, a t-SNE dimension reduction is applied, and the area is the total of different papers accessed by the corresponding group. The observed overlapping is illustrative and represents possible interconnections between them.

In another example, YP is recognized as improving symptoms and positive physiological effects in adults with rheumatic diseases. Yoga physical exercises have also improved functionality, pain, and quality of life in children with rheumatic diseases – Enthesitis-Related Arthritis (ERA) [34].

**3.2.2. Yoga affects the body and physiology.** The second dimension in this literature, characterised by the group II - Brain/Respiratory/Heart, comprises four strongly connected subgroups (A. Respiratory, B. Metabolic and Cardiovascular, C. Biomarkers, and D. Neurological), distributed into thirty-five LOGEXPs, with 4,749 articles and 988 title matches. The LOGEXP with the most significant number of articles is "BODY" (LOGEXP 73), followed by "CARDI | HEART" (LOGEXP 43), and "HYPERTENS | BLOOD | ARTERIAL & PRESSURE" (LOGEXP 75). The LOGEXP with the most significant number of titles is "CARDI | HEART" (LOGEXP 43), followed by "DIABET" (LOGEXP 84). The following aspects are correlated under the hierarchical structure depicted in Fig 3: Subgroups A and D are associated with respiratory and

neurological processes. In contrast, subgroups B are linked to metabolic and cardiovascular functions. Subgroup C is concerned with biomarkers. The subsequent sections will present it in detail.

The subgroups Respiratory (A) and Neurological (D), which pertain to respiration and the nervous system, respectively, are hierarchically more basal blocks and are widely connected to the YP in dimension I (Yoga as a physical and mental exercise) compared to other body aspects, suggesting the need for component "attention" to explain yoga integration, its derivative benefits, and the close relation of respiration as a mediator between practice and body. The use of breathing techniques to achieve meditation or relaxation is a common theme throughout the literature on yoga. The same techniques utilized for meditation in traditional YP effectively promote relaxation and enhance attention focus [36,37]. Self-perception, or awareness, represents the practitioner's active posture during exercises and links YP and nervous system functioning. The concept of awareness understood as 'being present', is also employed to enhance the capacity to sustain attention [5,9].

Subgroup D (Neurological) focuses on articles that address YP's neurological and brain-related aspects. Some reports present results of electroencephalography (EEG) and neuroimaging studies targeting the brain in the context of YP [36,37]. Others interpret, from a neuroscience point of view, the principles that may support the benefits the YP entails [5,9]. Moreover, other studies have examined biochemical markers in stress patients to elucidate the relationship between YP and the regulation of sympathetic and parasympathetic activities [38]. Recently, this theme has strongly emerged, and the volume of evidence points to this hypothesis as a key to understanding this group, with implications for all others. The literature indicates that an increase in parasympathetic activity and a decrease in sympathetic activity are associated with reduced stress levels [39] and improved cardiovascular function [40]. These, in turn, are related to many diseases, including diabetes (LOGEXP43 and 84, which correspond to "CARDI | HEART" and "DIABET", respectively), which also appear in this group. The autonomic nervous system (ANS) regulation is typically measured through Heart Rate Variability (HRV), a metric that deserves attention. The predominance of the High-Frequency component (HRV-HF) is indicative of heightened parasympathetic activity. Some studies employ HRV measures to compare experienced practitioners' outcomes with beginners or non-practitioners [39,41,42].

About subgroup A (Respiratory), the search conducted using the LOGEXPs"ASTHMA" and "PULMON | PNEUM", which yielded LOGEXP 81 and 32, has been retained in the context of breath (LOGEXP 81 - "ASTHMA") in the diagram presented in Fig 3. The regulation of ANS and the cardiovascular system is also affected by breath control learned by yoga practitioners [43,44]. Many articles on the cardiovascular system address cardiovascular diseases and their risk factors [45–47]. Moreover, risk factors are frequently the focus of studies examining hypertension, diet, and overweight. Other studies examine the regulation of the cardiovascular system by investigating its interactions with the parasympathetic system [43,48,49] (Fig 5).

Subgroup B (Metabolic and Cardiovascular) is strongly associated with metabolic and cardiovascular aspects. The autonomic nervous system is represented in the subgroup by the LOGEXPs "PATHETIC", "AUTONOMIC" and "VAGAL" (LOGEXP 98, 99, and 100, respectively - Fig 3 and 5). A comprehensive understanding of the ANS and its relationships in dimension II is essential for thoroughly examining the yoga-related literature. Notwithstanding, the LOGEXP 'HEMODYNAM' (LOGEXP 77) encompasses only 18 documents. Nevertheless, the Data Mining Model demonstrated the capacity to infer implicit relations, as evidenced by the association of yoga with hemodynamic adaptations and parasympathetic activities [43], as well as the other LOGEXPs within subgroup B (Fig 3).

Notably, several articles have highlighted additional beneficial effects of YP [50–52], although the necessity for further research is frequently emphasized. In this context, the literature reveals connections between Diabetes, Metabolic Syndrome, and even between these and Hypothyroidism. For example, the LOGEXP 'DIABET' (LOGEXP 84) yielded 154 articles on the relationship between yoga and diabetes. Of these, the LOGEXP 'DIABET' was present in the title of 104 articles. In addition to highlighting the advantages of yoga for individuals with diabetes, the literature, particularly of group II, offers perspectives on the physiological aspects of YP.

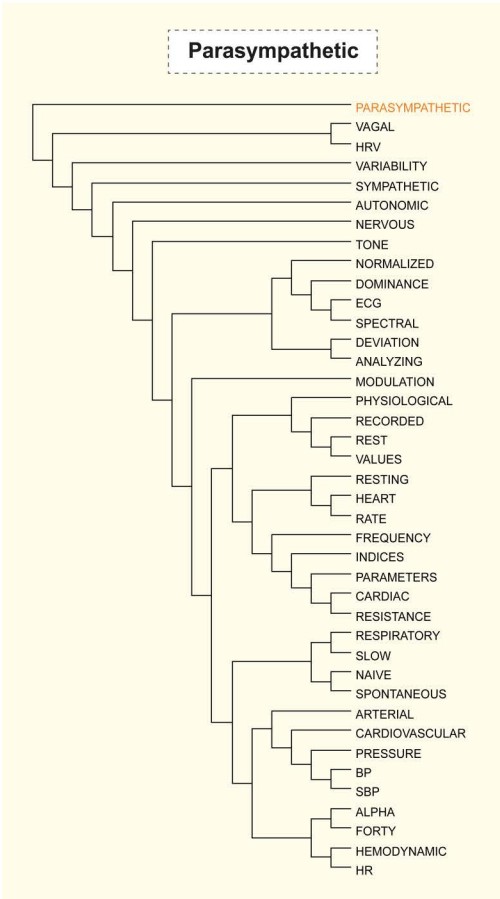

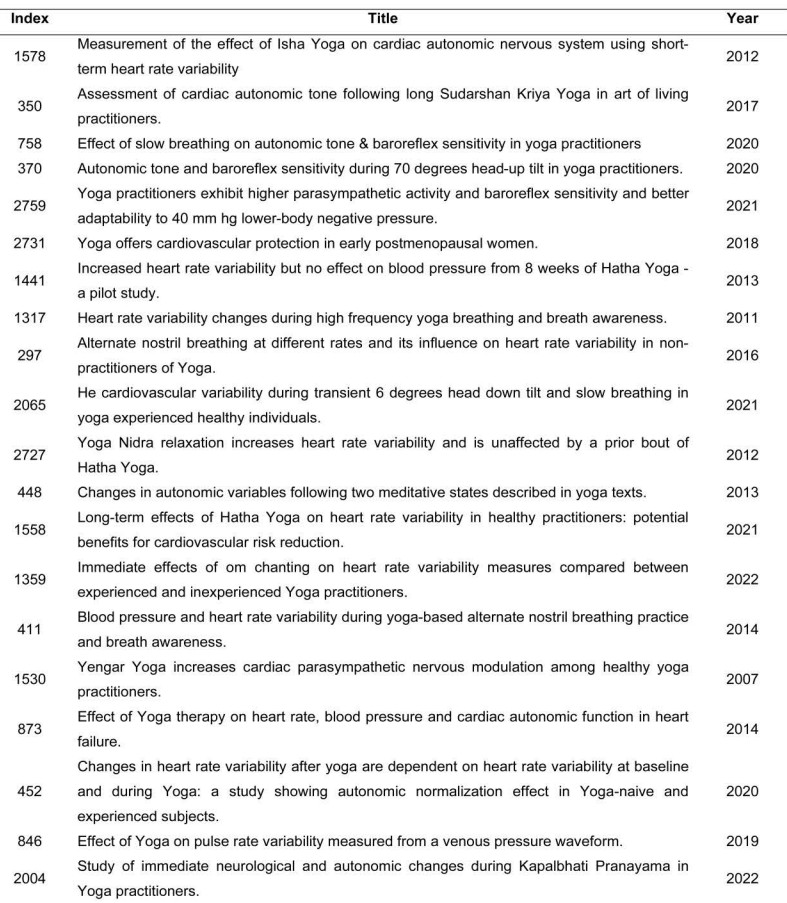

**Fig 5. Dimensional Scatterplot Visualization of Word Proximity.** It is possible to obtain the closest words or studies within the same vector space from a vectorized word. From the word 'PARASYMPATHETIC' as an example, we can receive: **A.** phylogenetic tree associating the word with the 40 closest words according to Yoga literature and **B.** list of the closest papers to the word. Note that the tree shows words related to the subject, such as 'NERVOUS', 'AUTONOMIC', and physiological characteristics modulated by the PARASYMPATHETIC system, such as 'HEART', 'RATE', 'CARDIAC', 'PRESSURE', 'HEMODYNAMIC', related to the cardiovascular system, and 'RESPIRATORY'. Similarly, the listed papers present the cardio-respiratory and autonomic nervous systems as the central subject of the study. The references for articles are: [39,41,43,48,49,62–76].

The subgroup C, represented by Biomarkers, comprises studies that present measurable markers to assess the effect of yoga (physical, biochemical, and genetic), demonstrating that YP acts on the various body systems, including the regulation of the immune system [1]. The studies revealed a pattern for regulating pro-inflammatory markers associated with YP, as evidenced by a reduction in IL-1beta, IL-6, and TNF-alpha (pro-inflammatory cytokines). Furthermore, the beneficial impact on hypertension may not be immediate, suggesting a sustained effect that the practice dosage may influence.

In this study, the LOGEXPs 'INFLAMM' and 'IMMUN' (LOGEXP 95 and 36) are employed to observe the effects on the sympathetic nervous system on immune-related parameters described. More recent studies, as outlined in this literature, corroborate the anti-inflammatory properties of YP and identify additional markers [53,54]. Other studies have investigated YP's potential to influence cellular aging reduction [55,56] positively. In these studies, telomerase and chromosome length have been proposed as indicative measures. Additionally, studies have examined the potential of YP to reduce oxidative

stress-related makers [55,57,58]. Other markers that are positively correlated with YP include the neuroplasticity marker Brain-Derived Neurotrophic Factor (BDNF) [59–61] and Gamma-Aminobutyric-Acid (GABA), which is present at low levels in patients with severe depression.

In this context, we presented a selection of markers whose relationship with yoga practice has been significantly established, representing just a tiny sample of the many such markers that have been identified. Table 2, adapted from Varambally and co-authors (2019) [3], lists additional features. Upon analysis of the results, it became evident that many of these markers exhibited a correlation, which suggests the potential for a standard to be applied to evaluate generalized YP results. However, this proposition requires further investigation through dedicated studies. Unfortunately, there is currently no precise association in the literature between each physiological effect and the various styles of yoga, given that the practice is treated generically in the analyses.

The table shows important markers that integrate with the other studies surveyed. Yoga is effective in therapy for psychiatric disorders. Adapted from [3].

**3.2.3. Psychosocial effects and theory of Yoga.** The third dimension encompasses the psychosocial effects and theory of Yoga, which correspond to the groups: Work Ability – School (III), Traditions theory (IV), Psychiatric disorder Eat - Substance (V), and Stress - Anxiety - Depression (VI) (Fig 3). Among these groups, the VI (Stress, Anxiety and Depression) contains the most significant number of LOGEXP, as well as the largest number of documents (nDcs and nTit).

The studies related to Work Ability – School (III) address specific social contexts, such as school and work environments, where implementing yoga (and Mindfulness)-based programs is evaluated. In the majority of cases, the results are obtained through qualitative testing. Such studies frequently involve groups of students at various levels [77–80] and teachers [81,82]. The objective is to ascertain whether YP can derive benefits concerning stress, anxiety, depression, resilience, self-esteem, compassion, social relationships, academic performance, mindfulness, flexibility, communication, empowerment, physical fitness, and self-regulation skills in a variety of forms (emotional, relaxation, mindfulness, and aggression). For instance, yoga has been demonstrated to assist in regulating emotions in students [80,83]. It has been shown to enhance the capacity of children on the autism spectrum to communicate their feelings [84,85]. Another example of the application of Yoga in an academic environment is its integration into anatomy teaching, wherein students utilize self-perception as a reference point for learning [86–88].

**Table 2. Mapped relations between Yoga Practice (YP) and positive neurobiological effects.**

| | Biomarkers | Effect Expected | Observable Effects |
|---|---|---|---|
| Neurotransmitters | ↑GABA, ↑Oxytocin, ↕Dopamine, ↑β-endorphins, ↑Acetylcholine, ↓Catecholamines | Modulation of Psychopathology | ↓ depression, ↓anxiety, mood improvement, ↑well- being |
| Inflammation and Oxidative stress Neurotrophic factors | ↑NF-kβ, ↑IRF, ↓IL-6, ↓TNF-α, ↑Adiponectin, ↑SOD, ↑Catalase, ↓PLA2, ↓Lipid Peroxidation, ↓Protein oxidation ↑BDNF, ↑Telomere length, Telomerase activity | ↑Neuroprotection† ↑Neuroplasticity† | |
| HPA axis | ↓Cortisol | Modulation of ANS | ↓ anxiety |

† Both promote ↑Cortical thickness and ↑Hippocampal volume.

↑ - indicates increased expression;

↓ - indicates a reduction in expression;

↕ - indicates variation in expression.

Similarly, there are studies in other settings, including pregnancy groups, healthcare facilities, and vulnerable communities. In addition to assessing the direct benefits to practitioners, the feasibility and adherence to the programs are also evaluated [89–91]. It is also noteworthy that the beneficial effects of yoga practice persist even after the practice is discontinued [92,93].

Group IV articles (Traditional Theory) establish theoretical links between yoga and science or philosophy, including yoga theory. Such approaches include traditional YP knowledge, relationships with contemporary science, frameworks describing YP, and socio-economic aspects of yoga. The concepts of self-image and embodiment are prominent in theoretical yoga studies in psychology. Some examples include studies related to obesity [8,11,93,94]. However, our analysis found elements of embodied cognition theory that can support elaborating new frameworks and strategies for designing studies on Yoga. These studies range from analyzing the social contexts of practitioners [94,95] to the relevance of the benefits of self-care based on yoga techniques to accessibility [96], including yoga in work settings and interpersonal relationships [97,98], and the ethical aspects present in the traditional principles of yoga [99].

The theoretical studies of yoga in neuroscience also represent this group. The recent emergence of studies aims to elucidate the mechanisms underlying the benefits of yoga practice. Studies relate YP with the Polyvagal Theory [9] and address the relevance of the association between yoga practice and the regulation of the autonomous nervous system, which occurs mainly by improving the tone of the Parasympathetic Nervous System [5]. The scientific studies that seek to substantiate the principles of YP are relevant in the current scenario, where the literature derives from different contexts and diverse approaches, which are essential to a better understanding and exploration of yoga techniques.

Nevertheless, new contributions with a multidisciplinary scope can integrate knowledge from the field of yoga. Schmalzl and co-workers (2015) propose three components when describing yoga practice: 'POSES', 'ATTENTION', and 'BREATHING' [5]. These three active forms of involvement, as evidenced in the literature, are integral to the efficacy of yoga practice. However, when these elements work together, they can yield enduring benefits analogous to those associated with another component: 'LEARNING'. Therefore, we propose that new structures for yoga be devised systematically, with contemplation of learning as one of its fundamental principles.

Group V (Psychiatric Disorder Eat-Substance) approaches yoga as a complementary therapy targeting psychiatric disorders, including eating and addictive disorders. The articles in this group highlight the evolution of yoga research in psychiatry [3]. Initially, yoga studies aimed only at the general well-being of practitioners, but later, the approach focused on treating more formalized aspects, considering measurable neurobiological effects relevant to medical practice. It also highlighted relationships that indicated important markers, mentioning the direction of the observed effect and its relationship to YP – positive aspects are expected in interventions. The diagram in Table 2 summarizes how to integrate this topic with others in this study, so we considered it essential to adapt and incorporate it. In addition, many of these markers have already been presented, reinforcing the advanced possibility that YP-related characteristics also correlate analogously. By following this principle, we can trace the possible relationships by traversing the evidence provided by the markers on the various research fronts and infer new potential relationships. This effort is beyond the scope of this study, but we believe it is a promising avenue for further studies.

Group VI (Stress - Anxiety - Depression), the last of the third dimension, is the group with the emphasis on psychology, where affective, mood, social relations, overload, and trauma issues are treated. In this group, the term with the most significant number of articles is "HEALTH" (LOGEXP 14), followed by "PHYSICAL" (LOGEXP 16) and "BENEF" (LOGEXP 87). The LOGEXP with the largest number of titles is "HEALTH" (LOGEXP 14), followed by "STRESS & ~OXIDATIVE" (LOGEXP 9) and "ANXIET | DEPRESS " (LOGEXP 10). Some studies have demonstrated that yoga is a potential adjunct in therapeutic processes or even a possible substitute in some cases, with advantages related to accessibility [100–102]. In the yoga literature, the term "PSYCHOL" (LOGEXP 113) is mainly associated with the LOGEXPs "ANXIET | DEPRESS" and "STRESS & ~OXIDATIVE", which account for more than 25% of the documents. These numbers show that thematic aspects are present in almost all the literature. The term "MINDFULNESS" (LOGEXP 12) strongly relates to the group

characteristics but presents different connotations. Sometimes, it means the present state of the present mind (or the ability to achieve it) [103], and sometimes figures as a technique to be used in association with yoga [104]. Several studies show the benefits of YP in groups with post-traumatic stress disorder (PTSD), including improving mindfulness in veterans [89,105] and reducing symptoms in women with PTSD [106,107]. Other studies associate lower cortisol levels with YP, reduced stress, and other mood-related disorders. This finding makes us consider including mood-related issues and other psychological aspects in the relationships with markers, which may open new research frontiers.

### 3.2.4. The Therapeutic Contribution of Yoga.

The fourth dimension englobes the therapeutic contributions of Yoga, which correspond to the following groups: Joints (VII), Feasibility safe (VIII), and Diseases (IX) (Fig 3). The group VII (Joints) is related to studies on the benefits of yoga on the neuromuscular system, such as the gain of mobility, gait and balance. These last three groups make up the meaning of the word Therapeutic because the main characteristic related to this dimension is the approach of YP to people in fragile situations.

Studies in this group include measuring ankle range of motion during YP to establish a parameter that can help understand the potential effects of YP on recovery from injury or surgery. For instance, yoga was applied in a case study to an individual with incomplete spinal cord injury, improving balance, flexibility, muscle strength, and performance on functional goals [108]. Another study evaluated the effect of a yoga program to minimize knee adduction moments in women with knee osteoarthritis, and the results showed improvements in pain reduction, strength, and mobility [109]. In this context, research described how a yogasana posture intervention affected balance performance in individuals with Peripheral Diabetic Neuropathy (NPD) [110]. Yoga was more effective than conventional exercise in improving static and dynamic balance in all standing balance variables and reducing fear of falling in individuals with NPD. Other related researches include topics such as physical function [111,112], musculoskeletal rehabilitation, mobility [113–115], balance [114,116], gait [114], and flexibility [115].

In the literature, particularly within group VIII (Feasibility Safe), studies have assessed the viability of integrating individuals with special needs into yoga groups, focusing on the potential personal and social benefits. Numerous studies have focused on groups of cancer patients [117,118] or those in treatment [119–121]. Other papers address groups of individuals experiencing chronic pain who require therapeutic practices [122,123]. The expected outcomes include improved sleep quality, mood, anxiety, depression, stress reduction, fatigue alleviation, enhanced physical and social functionality, and reduced overall suffering. However, the studies on this topic are not solely concerned with the well-being of the participants. They also demonstrate that Yoga-based therapies can play a significant role in addressing the shortcomings of conventional methods. The program's viability is evaluated based on effectiveness, recruitment, adherence, participation, safety, and satisfaction criteria. The conscious and focused movement, which provides personal and active involvement of those involved in the YP, may potentially reduce the risk of injury and adverse effects, which may serve as motivating factors for these studies.

Despite the considerable heterogeneity of diseases and syndromes encompassed by group IX (Diseases), the group's cohesion makes us hypothesize a tendency towards common elements in the disease literature, which may indicate new possible semantic correlations. The articles describe a distinctive quality of YP with outcomes in LOGEXPs of pain and suffering relief associated with the diseases. The literature indicates that yoga interventions can be a productive method for rehabilitating individuals with acquired brain injuries, such as stroke, and neurodegenerative diseases, including Parkinson's and Multiple Sclerosis. The literature supports this promising role of benefits to cardiovascular rehabilitation [124–127]. The primary results reported include improvements in cognition, mood, and stress reduction; increase in self-efficacy; motivation for physical activity; improvement in quality of life; better subjective perception of cardiac health; tendency to improve left ventricular systolic function; and improvement in motor function in balance. Concerning balance, a study [116] indicates that individuals who engage in yoga for longer demonstrate superior balance control. This finding substantiates the notion that yoga practice facilitates the formation of learning memories and subsequent learning effects.

While some articles highlight the pivotal role of yoga in system rehabilitation, the studies of [128] and [129] substantiate the benefits of yoga practice in relieving osteoarthritis-related pain.

A review of the literature on "MIGRAIN | HEADACHE" (LOGEXP 59) revealed that yoga practice resulted in clinical improvements, including reducing the frequency of headaches and the need for medication. Moreover, increased vagal tone and decreased sympathetic activity improve cardiac autonomic balance [130].

The term "FIBROMY" (LOGEXP 58), which refers to fibromyalgia, is directly related to various types of pain and has a direct relationship with joint pain literature ("ARTHRIT" LOGEXP 83). Some studies have proposed that yoga and mindfulness practices can improve the manifestations of fibromyalgia, enhance functional abilities, and strengthen coping skills. Furthermore, they modulate abnormal pain processing associated with fibromyalgia, markedly enhancing heat pain tolerance and pressure pain threshold [131]. Moreover, there are reports advocating that yoga and meditation significantly improved stiffness symptoms (along with anxiety and depression), thereby enhancing overall well-being [132].

The literature also links pain with irritable bowel syndrome (IBS), particularly highlighting symptoms such as abdominal discomfort, diarrhea, and constipation, with emotional stress exacerbating these symptoms [133,134]. Studies suggest that yoga may be a beneficial adjunctive treatment for IBS [135]. Additionally, yoga has been shown to help relieve primary dysmenorrhea and improve physical fitness and quality of life [136]. Research also indicates that yoga positively impacts the quality of life and balance in post-menopausal osteoporosis and contributes to overall physical and psychological well-being in women during the perimenopausal period [137,138].

Studies related to Yoga practice and central nervous system disorders also reported that yoga practices can be safe and effective for quality of life, balance ability, gait pattern, and joint flexion [6,139]. However, other articles claim that due to the heterogeneity of yoga practices and the lack of protocols or outcome indicators for practices targeting specific audiences, there is insufficient evidence confirming the benefit of yoga compared to other physical exercises. Literature still shows positive aspects of mind-body yoga, meditation, and relaxation techniques for individuals with chronic and debilitating neurological diseases such as Multiple Sclerosis, Stroke, or Parkinson's.

We found several studies reporting Yoga as a practice of a safe nature. These YP characteristics stimulate the adhesion in yoga groups aiming for physical activity. In addition, people with physical, emotional, or psychological challenges resulting from diseases or central nervous system disorders are an essential part of this public. People have a natural capacity for adaptation that includes reconnecting the body's poorly explored and injured areas. We conjecture that Yoga techniques explore these mechanisms efficiently, enabling physical, emotional, and neurological benefits. Additionally, while, unfortunately, conventional exercise-based rehabilitation programs are usually expensive, yoga presents a safe and low-cost alternative.

Our hypothesis that yoga generates beneficial learning also applies to improving the effects of strength application and biomechanical organization of the body and movement. Individuals with healthy joints deliver their body mobility, stability, flexibility, balance, dexterity, and gait quality and can improve the effects of biomechanical strength. The organized body system helps maintain focus, enhances coordination, provides neurological reconnections (Fig 5B), and regulates muscle tone and strength, among other aspects. This health condition certainly implies benefits in regulating other systems as well. However, we did not find studies that focus on joints systemically as a criterion for assessing YP. We noticed room in the literature for contributions along these lines.

### 3.3. Yoga styles and the focus of literature

One challenge in the literature was associating the diverse array of yoga techniques with their various benefits. Many studies treat yoga as a singular phenomenon, which presents a challenge in establishing protocols due to the heterogeneity of yoga practices [3]. Some research and reviews acknowledge the diversity of practices but do not discuss or evaluate the effects of each one independently [1,134]. In turn, in our study, we observed some interesting relationships between

**Table 3.** *Diversity of yoga styles and fields of study related to each style.*

| Yoga style | nDocs | nTit | Literature |
|---|---|---|---|
| Ashtanga | 28 | 11 | Ashtanga yoga emphasizes psychological and spiritual aspects, demonstrating efficacy in alleviating distress related to trauma, anxiety disorders, and depression. Research indicates improvements in motivation, self-determination, and overall mental health. Studies also explore its philosophical, spiritual, and transcendental dimensions alongside psychiatric benefits. |
| Bikram | 39 | 20 | Hot Yoga, a variant of Hatha Yoga, is practiced at temperatures around 40°C and should be done at light to moderate intensity with proper hydration to prevent dehydration. Its objectives include enhancing cardiovascular health, reducing arterial stiffness, lowering blood pressure, and promoting vasodilation, benefits that can be observed even in older adults. Additionally, Hot yoga helps reduce body weight in obese individuals, increases glucose tolerance, improves physical fitness, reduces stress, and boosts muscle mass. |
| Hatha | 288 | 113 | Hatha yoga offers physical conditioning, enhancing strength, flexibility, endurance, and weight loss while promoting mindfulness and improving self-image. It has therapeutic applications in mental health, reducing stress and postpartum depression, and improving self-image in breast cancer patients. Hatha yoga is also effective in alleviating fibromyalgia symptoms and aiding injury recovery. Additionally, it benefits the old people by reducing stress and exhaustion, and providing therapeutic effects for age-related diseases like osteoarthritis. |
| Iyengar | 102 | 42 | Iyengar yoga has beneficial effects on the physical and mental well-being of individuals with rheumatoid arthritis, restless legs syndrome, breast cancer, irritable bowel syndrome, and chronic back pain. It alleviates symptoms such as pain, insomnia, and fatigue while reducing reliance on medication, depression, and anxiety. Additionally, it enhances emotional health and overall quality of life. |
| Kriya | 93 | 42 | Kriya yoga is particularly effective in reducing anxiety, depression (including cases of alcohol dependency), stress, and insomnia while also promoting emotional control. It emphasizes relaxation, mindfulness, breathing, and meditation. |
| Kundalini | 46 | 46 | Kundalini yoga is primarily used in psychological treatments for conditions like obsessive-compulsive disorder, post-traumatic stress disorder, generalized anxiety disorder, sleep disorders, and substance abuse. It enhances sleep quality, resilience, emotional control, and memory and serves as a second-line treatment for generalized anxiety disorder. Additionally, Kundalini yoga is employed as a preventative measure against neurodegenerative effects and cognitive decline associated with Alzheimer's disease. |
| Vinyasa | 27 | 6 | Vinyasa yoga is a moderate to intense practice that positively impacts cardiorespiratory and metabolic systems, enhancing energy expenditure and aerobic capacity. Additionally, it supports weight loss, aids in smoking cessation, and improves stress, depression, and mood. |

*Legend*: **nTit**: Number of Titles; **nDcs**: Number of Document hits recalled by the logical expression (LOGEXP) whole set in a dimension.

yoga styles and fields of study, presented in Table 3. It is, however, noteworthy that our aim is not to show each style's limitations or specific effects but rather to identify trends in the literature.

## 4. Discussion

Exploring topics related to yoga through vector representations and semantic investigations reveals that yoga has a scientific basis, addressing the first question (*Does Yoga possess a robust scientific foundation?*). The upward trend in research production indicates a broad and ongoing interest in scientifically studying and understanding various aspects of yoga. The substantial volume of selected literature (5,785 of which 2,929 contain "yoga" in the title) not only reflects a significant corpus of research dedicated to yoga but also points to a growth trend, as evidenced by the notable number of literature reviews (306 articles). All yoga papers mentioned here are indexed in the PubMed database. While questions regarding the inherent quality of each article were beyond the scope of this study, other research focusing on article quality would be valuable.

Vectors could effectively represent the documents, LOGEXP, and words, preserving their semantic relationships, as shown in Fig 6A. To systematically analyze this body of research. We designed a set of 113 textual expressions (LOGEXP) for searches (in supplement S7) and constructed the global literature map. The LOGEXPs were encoded into vector representations to facilitate semantic searches within the 'YOGA' articles corpus. They were automatically organized hierarchically using a phylogenetic tree generation algorithm (Fig 3), highlighting the interconnectedness of various aspects and suggesting the robustness of Yoga's scientific underpinnings.

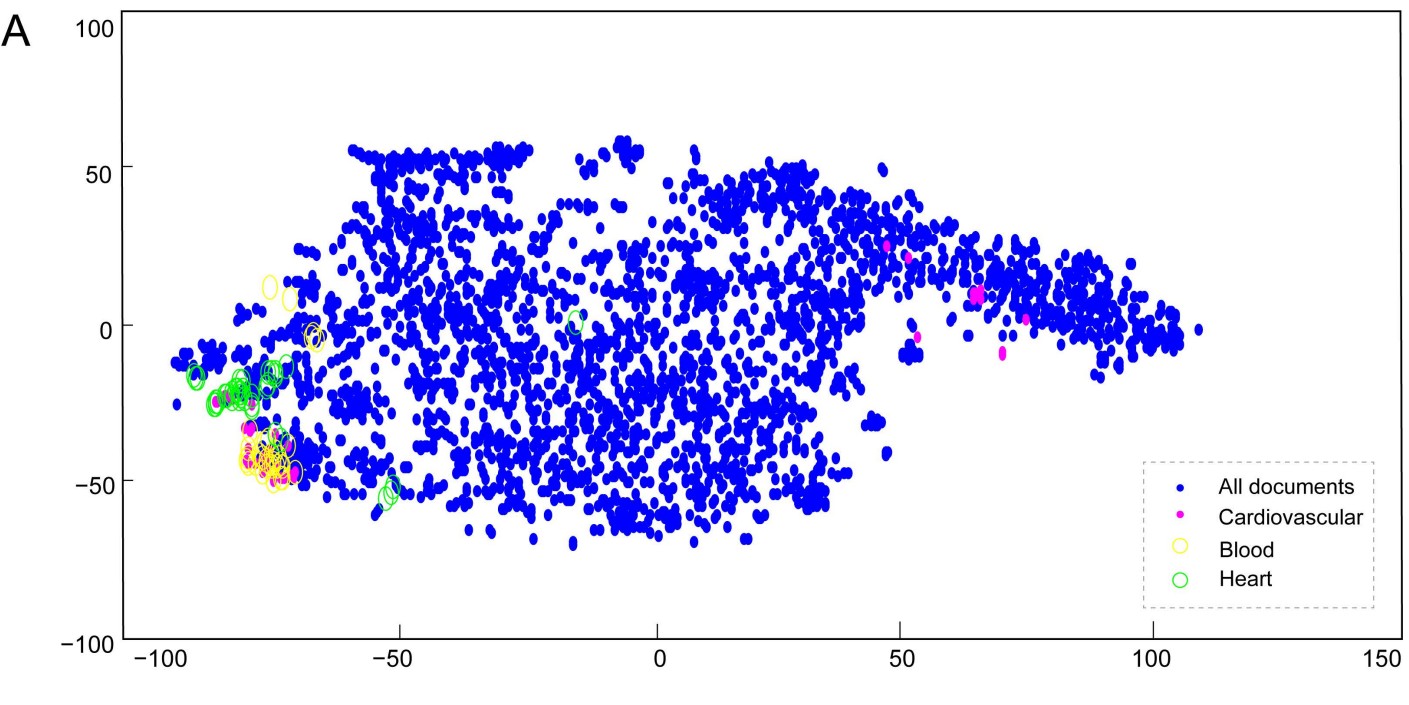

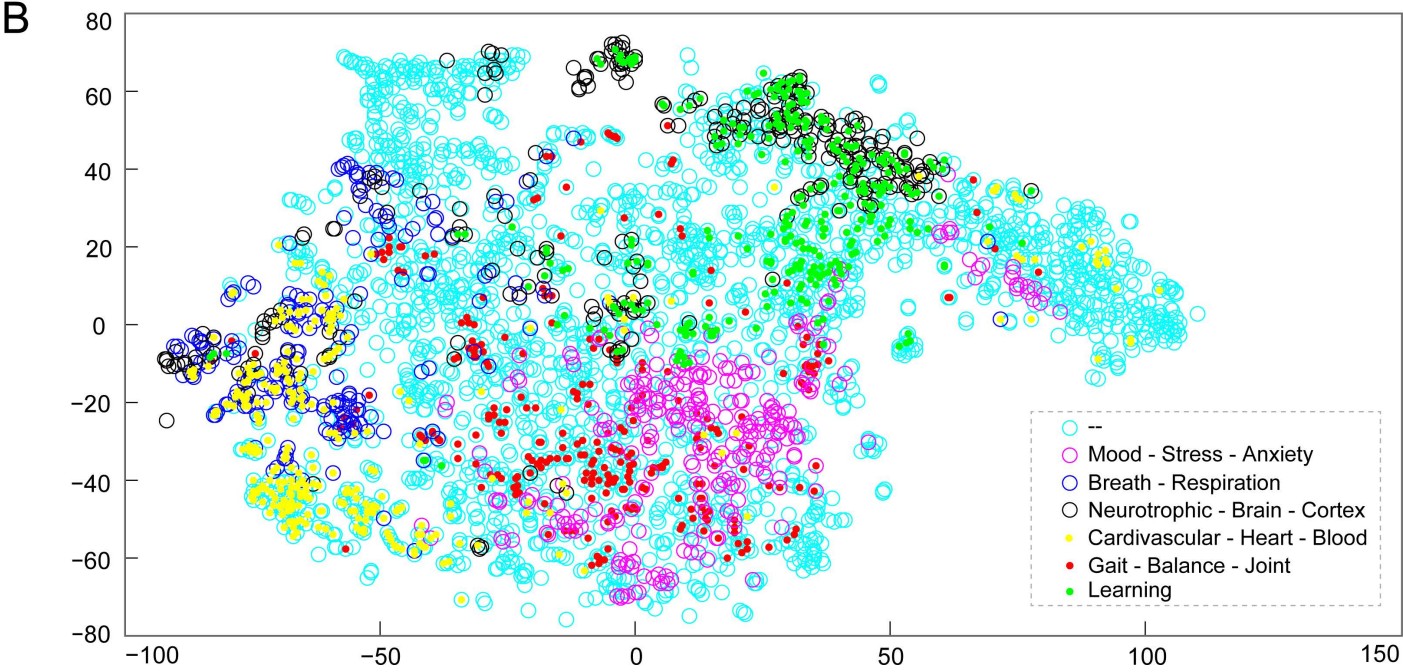

**Fig 6. t-SNE visualization of vectorized documents.** Each dot represents the 'TITLE+ABSTRACT' for each document. A. Semantic searches for 'HEART' (green), 'CARDIOVASCULAR' (magenta), and 'BLOOD' (yellow). The search considers 40 hits closest to each query. B. Distribution of 300 nearest hits for six semantics searches listed in the legend. Note that in B, the overlap between the topics of 'NEURO' (black) and 'LEARNING' (green) is visible, as well as between 'BREATH' (blue) and 'CARDIO' (yellow), and more subtly between 'MOOD' and 'BALANCE'.

## 4.1. Organizing yoga research

Building on this structured framework, the associations obtained permitted the literature to be discussed based on nine characteristic groups, with a line of integration between them. By delineating these nine coherent topics (Fig 3), we could elucidate and comprehend the fundamental aspects of yoga within the scientific literature. Four principal dimensions of yoga literature were identified, and the topics were classified according to their association with the most relevant dimension. The first dimension concerns the physical and mental aspects of yoga. The second dimension focuses on the physiological effects of yoga. The third dimension encompasses the psychosocial effects and theoretical aspects of yoga. The fourth dimension addresses the therapeutic contributions of yoga. The literature validated all observed relationships between words, with contextualization provided directly in related articles, which allowed us to interpret the contexts supplied by HTML-TM. The relationships found were coherent, and we present diagrams representing the context within the search for a word, facilitating a deeper examination of the literature. Despite its multifaceted nature, the literature on yoga is consistent, with essential connecting elements discernible even in disparate aspects.

In this context, the study of literature mapping enables us to gain insight into literature comprehension and, concurrently, to identify shortcomings and novel perspectives as we become more proficient in this field. The primary distinction between YP and conventional physical exercise is the degree of practitioner involvement, as evidenced by the literature's emphasis on the ATTENTION component. YP affects various body systems in a complex and integrated way, with notable neurological implications. The literature mapping elucidates the hierarchical interconnection between the practice and its bodily effects in subgroup D-Neurological of topic II (Brain/Respiratory/Heart). Furthermore, YP is an effective method for alleviating pain and suffering in patients with a range of physical and psychological impairments, and its practice is also considered safe. YP is also frequently addressed as an appropriate complementary therapy and a second-line treatment option.

Moreover, numerous studies explore the role of YP in mental health, including some of the most significant theoretical works. These studies suggest that the effects of YP may be generalized or extended to other areas of knowledge, with a particular focus on health, thereby addressing the second question: "*Can the effects of Yoga practice be generalized to other fields, particularly health?*". We propose possibly organizing biomarkers positively associated with YP to characterize its impact on health. Systematizing the use of these biomarkers is crucial for ensuring the reliability of research, as demonstrated by studies such as those conducted by Varambally and colleagues [3] (Table 2), which are especially relevant for psychological disorders.

Semantic searches can uncover connections among concepts that remain hidden with other approaches. For instance, as shown in Fig 5, the search LOGEXP 'PARASYMPATHETIC' links elements of the nervous, cardiac, and respiratory systems, illustrating the relationship established between these systems and yoga in the literature, indicating that an article might be relevant to a search even if the term itself does not appear explicitly in the text. For example, article [62], which is the most appropriate result for the search LOGEXP" PARASYMPATHIC", does not include the term explicitly, as illustrated in Fig 5).

In this review, we have drawn on the proposal of Schmalzl and co-workers, who describe YP as consolidated on three components: 'POSES', 'ATTENTION', and 'BREATHING' [5]. In our study, we propose including the 'LEARNING' component, which links the physical and mental axes of the practice. Here, 'LEARNING' encompasses a broad range of meanings, referring to the physiological or muscular configurations that become standardized in the practitioner's body and the cerebral aspects involved. While we recognize the importance of this topic, exploring it in depth is beyond the scope of our current study; nonetheless, it presents an exciting avenue for future research.

With many indications of benefits, Yoga is a promising source of support for treatments that can add positive effects to medicine, particularly in psychiatry. However, the complexity of the area and the methodological limitations of the studies make it difficult to make safe prescriptions in medical procedures and practices [140]. Many studies fail to be conclusive despite often providing positive results. We observed that the literature is more assertive when articles deal specifically

with Yoga concerning those that approach the theme in a joint context with other Complementary and Alternative Medicines (CAM). Therefore, in the definition of the LOGEXP, we considered only articles that explicitly contain the term 'YOGA' in the title. Furthermore, since we aim to understand yoga literature and not necessarily to substantiate Yoga-based therapies, we intend to understand the motivations of the studies and the relationships between them to point out new perspectives that contribute to the field.

## 4.2. Underexplored Aspects of Yoga Practice

Are there underexplored issues related to YP that require a more in-depth analysis? We were struck by the lack of representation in the literature of terms we consider essential in yoga, such as 'JOINTS' and 'LEARNING'. The articulations associated with the search LOGEXP 'JOINTS' or 'JOINT' (topic VII, LOGEXP 55) appear almost exclusively related to trauma or pathology studies. No studies have systematically examined the joints, evaluating the preventative and beneficial effects of joint health, such as mobility and flexibility, and other aspects mentioned in topic VII. The gain in joint quality with YP may be consistent with the known correlations and other elements in the literature and could, profitably, be further explored in future studies.

The term 'LEARNING' was excluded from the list of relevant words by the cut-off because the frequency of the term in yoga literature is lower than that observed among words in general use. Consequently, it is not in the HTML-TM search list. However, the term is present in the complete set of words and is associated with other terms and articles in our studies, but always with secondary and indirect connotations. Due to its low representation, we did not use 'LEARNING' in LOGEXP but other search terms involved in the concept. Two logical expressions have been developed, including "TEACHING", but in different contexts. The LOGEXP-8 (TEACH & {SCHOOL | STUDENTS}) targets relationships in the school environment, and LOGEXP-7 ({_TEACH | COACH} & ~{SCHOOL | STUDENT}) specifically targets teaching outside the school context and, consequently, Yoga learning/teaching, situated topic IV (Traditions-Theory).

The semantic relationship of LEARNING is, however, highlighted in Fig 6B, where we can see that the concept is closely related to the Neural-Brain aspects. The LEARNING aspect of YP receives little attention in studies, and this concept and its importance in achieving positive results with YP seems to be lacking in the literature. Fortunately, some studies indirectly corroborate the learning effect of YP in maintaining the results obtained, such as gaining resilience and physical, mental, emotional, cognitive, and other skills [141–145]. In this way, the literature also does not include aspects of learning as structuring elements for studying. To approach yoga as a field of research, we believe there is a need for studies that explicitly include learning as one of the foundations of yoga in the contemporary context. Aspects that could be addressed include the study of learning curves and the positive outcomes of YP, for instance.

The practice of yoga covers several styles (Table 3), and it is challenging to attribute specific benefits and practical distinctions to each style. As suggested in other literature, we chose to adopt the YP denomination as a general guide for this study. However, there is a significant gap in studies that systematically analyze and compare the benefits associated with each style of yoga. This lack of detailed research constrains our comprehension of each approach's specific characteristics and potential, underscoring the necessity for more comprehensive investigations in this domain.

## 4.3. Strengths, challenges, and future directions

This study is the first to provide a comprehensive qualitative overview of yoga, exploring the broad support of AI and MT techniques that facilitate research in the field. However, it still has some limitations, such as the analysis including only PubMed titles and abstracts, the lack of in-depth exploration of topics, and the non-inclusion of in-depth statistics, neither effect size calculation, nor meta-analysis. In addition, the strategy based on logical expressions (LOGEXP), created from literature reviews, broadened our analysis capacity; however, some relevant areas could be omitted in more specific domains. As a future perspective, we propose using full articles for more detailed analysis, improving the applied technique, expanding it to other fields of knowledge, and developing a platform to make this approach available to users.

Finally, what methods might researchers employ to situate themselves within the broader context of yoga literature and its constituent subfields? HTML-TM offers a framework for understanding the structure and relationships within the literature, facilitating navigation and contextualization within the subfields of yoga study (Table S12). In addition to the platforms through which the TM files can be accessed, we have incorporated our reference list, corresponding to the article's numbers in HTML-TM, which will enable readers to interact with the content of the article and the semantic models, thereby facilitating an in-depth analysis of the topic. More details on the content and use of HTML-TM can be found in the supplementary file. Ultimately, we believe that this material will assist yoga researchers in fostering multidisciplinary research networking, thereby contributing to the advancement of the field.

## Acknowledgments

The authors thank the colleagues of Artificial Intelligence Applied to Bioinformatics (AIBIA) and Graduate Program in Physical Exercise Medicine in Health Promotion of Federal University of Paraná, for their valuable discussions and contributions.

## Author contributions

**Conceptualization:** Rosangela Ieger-Raittz, Camilla Reginatto De Pierri, Roberto Tadeu Raittz.

**Data curation:** Rosangela Ieger-Raittz, Camilla Reginatto De Pierri, Roberto Tadeu Raittz.

**Formal analysis:** Rosangela Ieger-Raittz, Camilla Reginatto De Pierri, Camila Pereira Perico, Flavia de Fatima Costa, Jeroniza Nunes Marchaukoski, Roberto Tadeu Raittz.

**Investigation:** Rosangela Ieger-Raittz, Camilla Reginatto De Pierri, Camila Pereira Perico, Flavia de Fatima Costa, Roberto Tadeu Raittz.

**Methodology:** Rosangela Ieger-Raittz, Camilla Reginatto De Pierri, Camila Pereira Perico, Flavia de Fatima Costa, Diogo de Jesus Soares Machado, Roberto Tadeu Raittz.

**Project administration:** Camilla Reginatto De Pierri, Roberto Tadeu Raittz.

**Software:** Camilla Reginatto De Pierri, Diogo de Jesus Soares Machado, Roberto Tadeu Raittz.

**Supervision:** Roberto Tadeu Raittz.

**Validation:** Rosangela Ieger-Raittz, Camilla Reginatto De Pierri, Camila Pereira Perico, Flavia de Fatima Costa, Roberto Tadeu Raittz.

**Visualization:** Camilla Reginatto De Pierri, Camila Pereira Perico, Flavia de Fatima Costa, Roberto Tadeu Raittz.

**Writing – original draft:** Rosangela Ieger-Raittz, Camilla Reginatto De Pierri, Roberto Tadeu Raittz.

**Writing – review & editing:** Rosangela Ieger-Raittz, Camilla Reginatto De Pierri, Camila Pereira Perico, Flavia de Fatima Costa, Elisa Garbin Bana, Leonardo Vicenzi, Diogo de Jesus Soares Machado, Jeroniza Nunes Marchaukoski, Roberto Tadeu Raittz.

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
