## [Decision Letter · Decision Letter 0]

22 Jan 2025

PONE-D-24-47636What are we learning with Yoga? A text mining approach to LiteraturePLOS ONE

Dear Dr. Raittz,

Thank you for submitting your manuscript to PLOS ONE. After careful consideration, we feel that it has merit but does not fully meet PLOS ONE’s publication criteria as it currently stands. Therefore, we invite you to submit a revised version of the manuscript that addresses the points raised during the review process.

We look forward to receiving your revised manuscript.

Kind regards,

Kallol Kumar Bhattacharyya, MBBS MA PhD

Academic Editor

PLOS ONE

Journal Requirements:

2. Please note that PLOS ONE has specific guidelines on code sharing for submissions in which author-generated code underpins the findings in the manuscript. In these cases, we expect all author-generated code to be made available without restrictions upon publication of the work.

Please review our guidelines at https://journals.plos.org/plosone/s/materials-and-software-sharing#loc-sharing-code and ensure that your code is shared in a way that follows best practice and facilitates reproducibility and reuse.

4. Please note that your Data Availability Statement is currently missing the repository name. If your manuscript is accepted for publication, you will be asked to provide these details on a very short timeline. We therefore suggest that you provide this information now, though we will not hold up the peer review process if you are unable.

6. We are unable to open your Supporting Information file “supplemental_files.zip”. Please kindly revise as necessary and re-upload.

**Additional Editor Comments:**

As you can see that the reviewers have suggested some minor edits in the manuscript; therefore, I'd like you to carefully read and modify the manuscript accordingly. 

Reviewers' comments:

Reviewer's Responses to Questions

**Comments to the Author**

1. Is the manuscript technically sound, and do the data support the conclusions?

Reviewer #1: Partly

Reviewer #2: Partly

2. Has the statistical analysis been performed appropriately and rigorously? 

Reviewer #1: Yes

Reviewer #2: I Don't Know

3. Have the authors made all data underlying the findings in their manuscript fully available?

Reviewer #1: Yes

Reviewer #2: Yes

4. Is the manuscript presented in an intelligible fashion and written in standard English?

Reviewer #1: Yes

Reviewer #2: Yes

5. Review Comments to the Author

Reviewer #1: Thank you for the opportunity to review this interesting study and its methodology. I found the approach to be engaging and well-conceived. However, I would like to provide a few suggestions to enhance clarity and readability:

Abstract: The conclusion in the abstract appears somewhat vague. I would encourage you to provide a more precise and actionable takeaway to better communicate the key findings of the study.

Headings: In the "Background" section, I recommend using a more descriptive heading that directly reflects the content of the section. This will help guide readers more effectively through the material.

Results Section: To improve clarity, please consider structuring the results section more explicitly. Rather than referring to subgroups and groups using letters, it would be helpful to use descriptive names (e.g., "Respiration Group (A)"). This will make it easier for readers to follow and interpret your findings.

Tables and Formatting: I recommend reviewing the APA guidelines for formatting tables and for italicizing statistical abbreviations. Ensuring consistency and adherence to APA standards will improve the overall presentation of your work.

Clearer Presentation of Groups: A clearer and more detailed presentation of the groups, including their definitions and distinctions, will significantly enhance the reader’s ability to engage with and understand your findings. I would e.g. appreciate some figure to visualize all groups.

I hope these suggestions are helpful as you refine your manuscript. Thank you again for sharing your research.

Reviewer #2: First and foremost, I would like to appreciate the authors’ positive attitude towards the review process and constant efforts in revising and giving the manuscript a better shape. Their thorough attentiveness in analysing the descriptions and modification in the manuscript seems to be commendable. Their commitment for improving each aspect of the work from little details to extensive formatting seems praiseworthy. Now the manuscript provides a clearer and thorough vision. It can be said that the authors tried to provide an improved quality of the manuscript. However, there are some observations that needs attention-

• The manuscript explores research trends in Yoga, where the methodological approach for literature mining is majorly highlighted to develop a comprehensive framework for existing literature on yoga, facilitating multidisciplinary collaboration and bringing new light to relevant aspects. Thus, the title of the manuscript need to be re-framed indicating the methodology (literature mining approach) for describing Yoga.

• Define abbreviation e.g. BRET upon first appearance in the text.

• Authors have provided sub-sections like- Introduction and Background separately that needs to be improvised and re-organized following the IMRAD format.

• The basic queries of the manuscripts are to understand Yoga’s scientific foundation; the effects of Yoga practice be generalized to other fields, particularly health; identify underexplored topics within Yoga practice etc. However, it is difficult to get the direct corroboration with specific citations of the text mining results.

• Strengths and limitations of the study will help in improving the manuscript.

6. PLOS authors have the option to publish the peer review history of their article (what does this mean? ). If published, this will include your full peer review and any attached files.

**Do you want your identity to be public for this peer review?** For information about this choice, including consent withdrawal, please see our Privacy Policy .

Reviewer #1: No

Reviewer #2: No

---

## [Author Response · Author response to Decision Letter 1]

17 Feb 2025

We want to thank the reviewers for their excellent revisions, which have helped us to improve the manuscript in a general and definitive way. The feedback allowed us to improve its clarity and make it more informative for the community. We are sincerely grateful.

We are resubmitting the manuscript "What We are Learning with Yoga: A Text Mining Approach to Literature," with the title changed to "What are we learning with Yoga? Mapping the scientific literature on Yoga using a vector-text-mining approach", as requested by reviewer 2. We have considered all suggestions and incorporated the necessary revisions into the manuscript. Below, we provide detailed responses to each comment, and all modifications have been highlighted in yellow in the manuscript for clarity.

We have organized the supplementary material and addressed the editor's requests to avoid any delays in a potential publication. We point out that to make it easier to access the study's supplementary material, we migrated our supplementary files (S1 to S7 and the code developed to construct the HTML-TMs) to the Zenodo platform, with DOI: 10.5281/zenodo.14763946 (https://zenodo.org/uploads/14763946), instead of the file "supplemental_files.zip" (see lines 330-332; 943-944). We have also added the code developed and used to obtain our results (S11_Codes.zip) as supplementary material at the same link, as requested.

Reviewer #1:

Thank you for the opportunity to review this interesting study and its methodology. I found the approach to be engaging and well-conceived. However, I would like to provide a few suggestions to enhance clarity and readability:

Abstract: The conclusion in the abstract appears somewhat vague. I would encourage you to provide a more precise and actionable takeaway to better communicate the key findings of the study.

Response: We have emphasized our main conclusion in the text of the abstract, which we believe reflects the most important findings of our study (lines 38-42).

Headings: In the "Background" section, I recommend using a more descriptive heading that directly reflects the content of the section. This will help guide readers more effectively through the material.

Response: We changed the section's title from 'Background' to 'Definitions and Technical Details' (line 105) to make it more transparent.

Results Section: To improve clarity, please consider structuring the results section more explicitly. Rather than referring to subgroups and groups using letters, it would be helpful to use descriptive names (e.g., "Respiration Group (A)"). This will make it easier for readers to follow and interpret your findings.

Response: To make the text easier to read and follow, we have added the full names of each group and sub-group throughout the text, along with the corresponding Roman numerals and letters.

Tables and Formatting: I recommend reviewing the APA guidelines for formatting tables and for italicizing statistical abbreviations. Ensuring consistency and adherence to APA standards will improve the overall presentation of your work.

Response: As requested, we changed all the tables to APA format and presented them in a standardized way.

Clearer Presentation of Groups: A clearer and more detailed presentation of the groups, including their definitions and distinctions, will significantly enhance the reader's ability to engage with and understand your findings. I would e.g. appreciate some figure to visualize all groups.

I hope these suggestions are helpful as you refine your manuscript. Thank you again for sharing your research.

Response: We have restructured Table 1, which originally listed the dimensions and groups, to include subgroups (A, B, C, and D) and briefly describe each group. Further, we have incorporated the number of hits, unique occurrences, and coverage in Table 1, making the overlap between the defined groups more explicit. These additions improve clarity and help readers better understand the context of the yoga literature. Additionally, we have created Figure 4 to more effectively illustrate the overlap between the groups (lines 404–405, 411–41). Figure 3 also provides an overview of the groups, distinguished by color and their respective logical expressions.

Reviewer #2:

First and foremost, I would like to appreciate the authors' positive attitude towards the review process and constant efforts in revising and giving the manuscript a better shape. Their thorough attentiveness in analysing the descriptions and modification in the manuscript seems to be commendable. Their commitment for improving each aspect of the work from little details to extensive formatting seems praiseworthy. Now the manuscript provides a clearer and thorough vision. It can be said that the authors tried to provide an improved quality of the manuscript. However, there are some observations that needs attention-

• The manuscript explores research trends in Yoga, where the methodological approach for literature mining is majorly highlighted to develop a comprehensive framework for existing literature on Yoga, facilitating multidisciplinary collaboration and bringing new light to relevant aspects. Thus, the title of the manuscript need to be re-framed indicating the methodology (literature mining approach) for describing Yoga.

Response: We changed the manuscript's title from "What We are Learning with Yoga: A Text Mining Approach to Literature" to "What are we learning with Yoga? Mapping the scientific literature on Yoga using a vector-text-mining approach", emphasizing our methodological approach.

• Define abbreviation e.g. BRET upon first appearance in the text.

Response: We have revised the acronyms in the document and corrected the passages where they were not defined when they first appeared.

• Authors have provided sub-sections like- Introduction and Background separately that needs to be improvised and re-organized following the IMRAD format.

Response: To incorporate the suggested IMRAD format, we have changed the 'Background' section to an introductory subsection, which we renamed 'Definitions and Technical Details' for clarity.

• The basic queries of the manuscripts are to understand Yoga's scientific foundation; the effects of Yoga practice be generalized to other fields, particularly health; identify underexplored topics within Yoga practice etc. However, it is difficult to get the direct corroboration with specific citations of the text mining results.

Response: To make it clearer what each stage of text mining contributes to obtaining our findings, we have drawn up Table S12 (placed as a supplement to the manuscript and deposited it on the Zenodo platform – DOI: 10.5281/zenodo.14763946). In this table, we clarify that the TM was a collaboration between the information presented by AI and human analysis, which helped refine the other. We modified our manuscript in lines 355-358 to present this addition. We would also point out that throughout the text, we have made other minor changes (highlighted in yellow) to emphasize the interpretation of the TM.

• Strengths and limitations of the study will help in improving the manuscript.

Response: Among the study's strengths, we emphasize that it provides a facilitator for the yoga researcher. We present an overview of Yoga for the first time. Our limitations relate to the exclusive use of titles and abstracts of studies on PubMed, as considerable literature may exist outside the platform. We did not carry out meta-analyses or deepen the analysis of the groups. Furthermore, our strategy may have omitted relevant areas, considering that we constructed the logical expressions (LOGEXP) from the survey of bibliographic reviews. Considering this, we have the prospect of using full articles in the mining, which would provide a deeper understanding of the field. We also aim to expand the technique, use the knowledge acquired in other areas of study, and offer the technique as a platform for users. In the discussion, we added a paragraph to clearly present the study's strengths, limitations, and perspectives. These are presented in lines 919-928.

We hope that the adjustments made, as suggested by the reviewers, have improved the manuscript's presentation and comprehensibility. Once again, thank the reviewers for their valuable feedback and contributions to help us improve the manuscript.

Kind regards,

Roberto Tadeu Raittz, Ph.D, Corresponding author

---

## [Decision Letter · Decision Letter 1]

11 Mar 2025

PONE-D-24-47636R1What are we learning with Yoga? Mapping the scientific literature on Yoga using a vector-text-mining approachPLOS ONE

Dear Dr. Raittz,

Thank you for submitting your manuscript to PLOS ONE. After careful consideration, we feel that it has merit but does not fully meet PLOS ONE’s publication criteria as it currently stands. Therefore, we invite you to submit a revised version of the manuscript that addresses the points raised during the review process.

 This is a great work; however, I only could secure one review from one of the original reviewers. At this point, I do not want to send it out to new reviewers. Therefore, I reviewed the manuscript and found most of the comments were perfectly answered. I would like to request to respond one reviewer's minor suggestions and submit a revised version. You are very close. 

We look forward to receiving your revised manuscript.

Kind regards,

Kallol Kumar Bhattacharyya, MBBS MA PhD

Academic Editor

PLOS ONE

Journal Requirements:

Reviewers' comments:

Reviewer's Responses to Questions

**Comments to the Author**

1. If the authors have adequately addressed your comments raised in a previous round of review and you feel that this manuscript is now acceptable for publication, you may indicate that here to bypass the “Comments to the Author” section, enter your conflict of interest statement in the “Confidential to Editor” section, and submit your "Accept" recommendation.

Reviewer #1: All comments have been addressed

2. Is the manuscript technically sound, and do the data support the conclusions?

Reviewer #1: Yes

3. Has the statistical analysis been performed appropriately and rigorously? 

Reviewer #1: I Don't Know

4. Have the authors made all data underlying the findings in their manuscript fully available?

Reviewer #1: Yes

5. Is the manuscript presented in an intelligible fashion and written in standard English?

Reviewer #1: Yes

6. Review Comments to the Author

Reviewer #1: Dear Authors,

Thank you for your work on this paper. I found it very interesting to read, and it is clear that a great deal of effort has gone into this research. Although I am not an expert on the methodology used, it appears to be sound and rigorous, making the paper a strong candidate for publication. I do, however, have a few minor suggestions that could improve clarity and readability:

Tables and Abbreviations

When using titles in tables, figures, or similar elements, please ensure that all abbreviations are explicitly defined. For example:

Table 2. Mapped Relations Between YP and Positive Neurobiological Effects (YP = Yoga Practice).

Effect Sizes

I would appreciate it if effect sizes were included wherever available. If they are not reported, please mention this explicitly to provide full transparency.

Table Formatting and Ordering

Table 3 does not follow APA style. Please revise it accordingly.

The order of elements in Table 3 and Appendix B seems somewhat arbitrary. Organizing them by the number of studies found, year of publication or alphabetically could improve readability. If a specific rationale for the current order exists, please clarify it.

Discussion Section Clarity

Some parts of the discussion are difficult to follow. For instance, the first sentence:

"Exploring topics related to yoga through vector representations and semantic investigations reveals that yoga has a scientific foundation, addressing the first question."

The phrase "the first question" is unclear. Readers should not have to search through the document to understand what is being referenced. Please consider rewording for greater clarity.

Headings in the Discussion Section

Adding subheadings, such as Strengths & Weaknesses, would help structure the discussion and make it easier to navigate.

Overall, I enjoyed reading the paper and appreciate the depth of analysis presented. With these minor refinements, I believe the manuscript will be even stronger. Thank you again for your work!

Best regards,

7. PLOS authors have the option to publish the peer review history of their article (what does this mean? ). If published, this will include your full peer review and any attached files.

**Do you want your identity to be public for this peer review?** For information about this choice, including consent withdrawal, please see our Privacy Policy .

Reviewer #1: No

---

## [Author Response · Author response to Decision Letter 2]

26 Mar 2025

Dear Kallol Kumar Bhattacharyya, MBBS MA PhD

Academic Editor of PLOS ONE,

We are resubmitting the manuscript “What are we learning with Yoga? Mapping the scientific literature on Yoga using a vector-text-mining approach” to respond to the additional requests made by reviewer 1. We have considered all the suggestions and incorporated the necessary revisions into the manuscript, highlighting them in yellow. Below, we provide detailed responses to each comment.

We thank the reviewers and editors. The changes have greatly improved the document during this revision process.

Reviewer #1:

Thank you for your work on this paper. I found it very interesting to read, and it is clear that a great deal of effort has gone into this research. Although I am not an expert on the methodology used, it appears to be sound and rigorous, making the paper a strong candidate for publication. I do, however, have a few minor suggestions that could improve clarity and readability:

We thank the reviewer for his suggestions and hope that we have met his expectations.

Tables and Abbreviations

When using titles in tables, figures, or similar elements, please ensure that all abbreviations are explicitly defined. For example:

Table 2. Mapped Relations Between YP and Positive Neurobiological Effects (YP = Yoga Practice).

Response: We have incorporated the full meaning of the acronyms into the figures' and tables' titles and captions. Modifications are highlighted in yellow (see lines 242-3,377,412,416,562,780-1).

Effect Sizes

I would appreciate it if effect sizes were included wherever available. If they are not reported, please mention this explicitly to provide full transparency.

Response: We appreciate the reviewer's suggestion about including effect sizes. However, it is important to note that the methodology proposed in this study does not involve hypothesis testing or association analyses since our focus was on text mining. The main objective was to contextualize and demonstrate the relevance of the articles in the field of yoga without going into statistical depth. As such, effect size analysis does not apply to the scope of this work. We suggest that readers consult the original articles for more detailed statistical information. To ensure greater clarity, we emphasized that our review prioritized a qualitative approach through text mining (lines 924,928). We hope this explanation clarifies our methodological approach.

Table Formatting and Ordering

Table 3 does not follow APA style. Please revise it accordingly.

The order of elements in Table 3 and Appendix B seems somewhat arbitrary. Organizing them by te number of studies found, year of publication or alphabetically could improve readability. If a specific rationale for the current order exists, please clarify it.

Response: We changed the order of the elements in Table 3, respecting the alphabetical order. We also formatted Table 3 to conform to APA format, inserting two columns with the number of documents and titles, making the information more straightforward to read.

We checked Supplementary Table S12, and the order is in line with the sequence in which the study was carried out, which is made clear in the text (lines 358-9). However, we could not identify what the comment about Appendix B refers to.

Discussion Section Clarity

Some parts of the discussion are difficult to follow. For instance, the first sentence:

"Exploring topics related to yoga through vector representations and semantic investigations reveals that yoga has a scientific foundation, addressing the first question."

The phrase "the first question" is unclear. Readers should not have to search through the document to understand what is being referenced. Please consider rewording for greater clarity.

Response: We appreciate the reviewer's suggestion, and for clarity, we have left questions 1 and 2, mentioned in the discussion, explicit (lines 787-8,837-8).

Headings in the Discussion Section

Adding subheadings, such as Strengths & Weaknesses, would help structure the discussion and make it easier to navigate.

Response: We have subdivided the discussion into three topics to make it easier to navigate the text. The subsections were 4.1 Organizing Yoga Research (line 805), 4.2 Underexplored Aspects of Yoga Practice (line 873), and 4.3 Strengths, Challenges, and Future Directions (line 925).

Best regards,

Roberto Tadeu Raittz, PhD corresponding author

---

## [Editor Report · Decision Letter 2]

28 Mar 2025

What are we learning with Yoga? Mapping the scientific literature on Yoga using a vector-text-mining approach

PONE-D-24-47636R2

Dear Dr. Raittz,

We’re pleased to inform you that your manuscript has been judged scientifically suitable for publication and will be formally accepted for publication once it meets all outstanding technical requirements.

Kind regards,

Kallol Kumar Bhattacharyya, MBBS MA PhD

Academic Editor

PLOS ONE
---

## [Editor Report · Acceptance letter]

PONE-D-24-47636R2

PLOS ONE

Dear Dr. Raittz,

I'm pleased to inform you that your manuscript has been deemed suitable for publication in PLOS ONE. Congratulations! Your manuscript is now being handed over to our production team.

Kind regards,

on behalf of

Dr. Kallol Kumar Bhattacharyya

Academic Editor

PLOS ONE